

# Aerosol effects modeling using an online coupling between the meteorological model WRF and the chemistry-transport model CHIMERE

Régis Briant[1,2], Paolo Tuccella[1], Adrien Deroubaix[1], Dmitry Khvorostyanov[1], Laurent Menut[1], Sylvain Mailler[1], and Solène Turquety[1]

[1]LMD, Laboratoire de météorologie dynamique, École Polytechnique, 91128 Palaiseau, FRANCE
[2]Climatic Change and Climate Impacts, Institute for Environmental Sciences, University of Geneva, Boulevard Carl-Vogt 66, CH-1205 Geneva, Switzerland

*Correspondence to:* Régis Briant, rbriant@lmd.polytechnique.fr

**Abstract.** The presence of airborne aerosols affects the meteorology as it induces a perturbation in the radiation budget, the number of cloud condensation nuclei and the cloud micro-physics. Those effects are difficult to model at regional scale as several distinct models are usually involved. In this paper, the coupling of the CHIMERE chemistry-transport model with the WRF meteorological model using the OASIS3-MCT coupler is presented. WRF meteorological fields along with CHIMERE
aerosol optical properties are exchanged through the coupler at a high frequency in order to model the aerosol direct and semi-direct effects. The WRF-CHIMERE online model has a higher computational burden than both models ran separately in offline mode (up to 42% higher). This is mainly due to some additional computations made within the models such as more frequent calls to meteorology treatment routines or calls to optical properties computations routines. On the other hand, the overall time required to perform the OASIS3-MCT exchanges is not significant compared to the total duration of the simulations. The
impact of the coupling is evaluated on a case study over Europe, northern Africa, Middle East and western Asia during the Summer 2012, through comparisons of the offline and two online simulations (with and without the aerosol optical properties feedback) to observations of temperature, Aerosol Optical Depth (AOD) and surface PM10 (particulate matter with diameters lower than 10µm) concentrations. Result shows that using the optical properties feedback induces a radiative forcing (average forcing of -4.8W.m$^{-2}$) which creates a perturbation in the average surface temperatures over desert areas (up to 2.6° locally)
along with an increase of both AOD and PM10 concentrations.

## 1 Introduction

Both the direct and semi-direct aerosol effects refer to the perturbation of the radiation budget induced by the presence of aerosol in the atmosphere along with the induced changes in the meteorology (e.g. surface temperature, wind velocity, cloud coverage) (Jacobson et al., 2007; Hansen et al., 1997). The indirect aerosol effects refer to changes in the number of cloud
condensation nuclei along with the induced perturbations within the cloud micro-physics, thus of the cloud albedo and precipitations (Jones et al., 1994). The aerosol effects processes are known to have a significant impact on meteorology and on





airborne aerosol concentrations (Yu et al., 2006). However aerosol effects are difficult to model precisely as studies focusing on chemistry and meteorology usually involve two distinct models. Hence, they are neglected by offline models, as they are not capable of taking aerosol feedbacks into account. Developing fully-coupled online models, able to accurately take aerosol effects into account is a major scientific challenge (Zhang, 2008).

Online modeling approach enables the possibility for several models to be run concurrently and allows them to communicate with each other. Thus, it creates the possibility of feedbacks modeling, as models may interact both ways at each time step. Online models coupling meteorological models and chemistry-transport models (CTMs) are increasingly used (Baklanov et al., 2014). Merging two models in order to form a unique model is one solution (e.g. WRF-CHEM Grell et al. (2005), CMCC-CESM-NEMO (Fogli and Iovino, 2014), IFS-ECWAM-NEMO Breivik et al. (2015)). With this method all variables are shared,

however once models are merged it may be difficult to make each model component evolve independently. This is an issue when several independent modeling teams are involved or when more than two models are coupled. Using an external coupler to handle the variable exchanges is an alternative. Each model is interfaced with the coupler, allowing them to retain their independent course of development. The coupler may perform some operations on the coupling fields, such as interpolations. This approach is also a manner of sharing new model developments among research groups while allowing each group to

continue to administrate their own model. This approach has been applied to several online coupling platform such as WRF-CMAQ (Wong et al., 2012), CNRM-CM5 (Voldoire et al., 2013) or MPI-ESM (Giorgetta et al., 2013; Jungclaus et al., 2013).

OASIS is a widely used external coupler developed by the CERFACS (Centre Européen de Recherche et de Formation Avancée en Calcul Scientifique, Toulouse, France) (Valcke et al., 2015). Several geoscience models such as ECHAM (Stevens et al., 2013), LMDz (Hourdin et al., 2006) or ORCHIDEE (Krinner et al., 2005), have been interfaced with OASIS and

therefore, the OASIS coupler is used in several online models, such as EC-earth (Sterl et al., 2012), TerrSysMP (Gasper et al., 2014), the Met Office Unified Model (Williams et al., 2015) or IPSL-CM5 (Dufresne et al., 2013).

Several online-coupled regional air quality models have been developed (Im et al., 2015) and many studies focused on the aerosol radiative impacts. Pérez et al. (2006) studied the interaction between mineral dust and solar radiation through the inclusion of mineral dust radiative effects within the regional atmospheric dust model DREAM (Nickovic et al., 2001).

The feedback attributed to mineral dust is negative with a 35-45% reduction of the aerosol optical depth (AOD) over the Mediterranean region during a major mineral dust outbreak. Vogel et al. (2009) used the fully online coupled model COSMO-ART over western Europe and showed that aerosol particles induce an average decrease of the 2 meter temperatures (0.1K over Germany). Han et al. (2012) showed that mineral dust particles induce a decrease up to $90W.m^{-2}$ of long-wave radiative forcing along with an increase of $40W.m^{-2}$ of short-wave radiative forcing, when using the online regional climate-chemistry-

aerosol model RIEMS-Chemaero over east Asia.

In Péré et al. (2011), aerosol radiative effects over Europe are evaluated using both the Weather Research and Forecasting (WRF) meteorological model (Skamarock et al., 2007) and the CHIMERE regional chemistry-transport model (Schmidt et al., 2001; Bessagnet et al., 2004; Menut et al., 2013). Results indicate that the presence of particles induces perturbations in both the solar radiation (radiative forcing at the bottom of the atmosphere of $-30W.m^{-2}$ to $-10W.m^{-2}$) and the near-surface

temperatures (decrease up to $0.30 \pm 0.06$ K). An offline coupling was made by forcing the WRF model with aerosol optical




properties computed from CHIMERE outputs. Initially, the CHIMERE model was forced by the WRF model itself, thereby implying the need for developing interactions between the two models. The WRF model was recently interfaced with the OASIS coupler (Valcke et al., 2015) and coupled online to the NEMO (Nucleus for European Modelling of the Ocean) ocean model (Madec, 2008) in order to better study air-sea interactions (Samson et al., 2014). On the other hand, recent developments

within the CHIMERE CTM, made for the CHIMERE2016a release, were related to the development of an online version of the CHIMERE model. These developments have been pursued leading to the creation of an OASIS interface within the CHIMERE model. A WRF-CHIMERE online coupling was created, allowing the two models to exchange fields at each main physical time step (i.e a few minutes), thus enabling the possibility of the aerosol effects modeling.

This paper presents the online coupling developments made within the CHIMERE model along with an evaluation study

of the direct and semi-direct aerosol effects using the WRF-CHIMERE online coupling. Section 2 focus on the CHIMERE-OASIS interface that was developed within the CHIMERE model along with the scheduling of the WRF-CHIMERE OASIS exchange operations. An evaluation test case along with model configurations are presented in Section 3. In Section 4, the computational performances of the WRF-CHIMERE online coupling are compared to the performances of both offline models. In addition, an estimation of the OASIS exchange burden is made. Case study simulations over the summer 2012 are evaluated

in Section 5. WRF and CHIMERE offline simulations are compared to two WRF-CHIMERE online simulations. In the first online simulation, the CHIMERE model is forced by the WRF model, without any feedback but at a higher rate than what is possible in offline mode. In the second online simulation aerosol optical properties are transferred from CHIMERE to WRF in order to take into account the direct and semi-direct aerosol effects. Simulated results are compared to temperatures, AOD and concentration measurements. Note that applications presented in this paper focus on the aerosol direct and semi-direct effects

only. The study of cloud-aerosol interactions is currently ongoing and shall be the focus of a forthcoming paper.

## 2  Development of the WRF-CHIMERE coupled version

The CHIMERE2016a release included preliminary technical changes for the development of an online coupled version of CHIMERE. CHIMERE preprocessors (for the calculation of emissions in particular) were included into its core. Indeed, in case of an online simulation not all input data are known, prior to the simulation, for the entire simulation period. In particular, in

case of a WRF-CHIMERE online coupling, meteorological fields are received at each time step of a simulation, thus preventing from the precomputation of meteorology-dependent variables such as mineral dust emission or biogenic emission fluxes. Furthermore, CHIMERE held a master/worker pattern where the master process performed all input/output operations. A more efficient pattern was implemented in which each worker performs parallel input/output operations, using the Parallel-Netcdf library (Li et al., 2003), without any master process.

Pursuing the development of an online version of CHIMERE in order to perform a WRF-CHIMERE coupling, more developments were made since the CHIMERE2016a release. These developments are described in Sections 2.1 to 2.4 and fulfill the implementation of an online coupled version of CHIMERE.



## 2.1 The CHISIS interface module

A Fortran module, called CHISIS (CHImere / oaSIS), that interfaces CHIMERE and OASIS was developed. This module gathers all calls to OASIS subroutines required by CHIMERE in order to exchange fields with another model. Furthermore, a reading routine of the OASIS configuration file (i.e. namcouple file), was included, thus allowing each model to be aware of coupling parameters (e.g. exchanged variable names, time steps, partitions, grids, models involved), leading to generic subroutines without any hard-coded information. Even though the CHISIS module was designed for CHIMERE, it does not contain any CHIMERE specific material, therefore it may be used in other models.

An interface module already exists in WRF as a WRF-NEMO coupling has already been implemented. However, WRF-NEMO exchanged variables were hard-coded within the WRF model making it difficult to adapt for a WRF-CHIMERE coupling. Thus, new compilation flags were added within the WRF code to distinguish the hard-coded WRF-NEMO coupling code from the additional WRF-CHIMERE coupling code.

## 2.2 OASIS configuration

The latest OASIS release, OASIS3-MCT, internally uses the Model Coupling Toolkit (MCT), developed by the Argonne National Laboratory in the USA, for parallel regridding and parallel distributed exchanges of the coupling fields (Larson et al., 2005; Jacob et al., 2005). To perform a WRF-CHIMERE coupling, the exchange of three-dimensional fields are required (e.g. temperature, wind velocity, pressure). A previous OASIS release, OASIS4, allows to exchange three-dimensional variables (Redler et al., 2010), however its code was too complex to evolve easily, thus OASIS developers decided to take a step back and use MCT with the OASIS3 release that do not includes the possibility of three-dimensional variable exchange. To overcome this issue, three-dimensional variables are decomposed into one-dimensional arrays prior to the exchange. Doing so makes it impossible for OASIS to perform a spatial interpolation between both model grids, as OASIS then considers one-dimensional unstructured arrays instead of spatial grids. Thus, both WRF and CHIMERE models need to be run on the same horizontal grid in online mode. WRF vertical grid may be used as it is not dependent of the sub-domain decomposition of each model.

Both WRF and CHIMERE codes are parallelized using a decomposition into sub-domains which may be different for both models. A OASIS partition is required to describe each point of each sub-domain of each model within the same global index space. The OASIS "points" partition was chosen as it allows to index each grid point separately, thus ensuring to preserve models sub-domain decomposition flexibility, unlike other partitions that require to index segments of points or rectangular regions of a domain.



## 2.3 OASIS exchange

### 2.3.1 Exchange from WRF to CHIMERE

In order to be run in offline mode, the CHIMERE model requires 28 meteorological variables at a hourly rate. In the offline version, these variables are read from WRF output files and include both two-dimensional variables (e.g. 10 meter wind velocities, surface pressure, 2 meter air temperature) and three-dimensional variables (e.g. base state pressure, cloud Water mixing ratio, water Vapor mixing ratio). CHIMERE performs a temporal interpolation between two sets of hourly WRF fields in order to compute species concentrations at every physical time step (i.e. a few minutes).

The WRF-CHIMERE online coupling enables the possibility to avoid these sub-hourly temporal interpolations. Indeed, even though WRF output files may be hourly, WRF computes meteorology with a finer time step that is defined in its configuration file. Therefore, in online mode, the CHIMERE physical time step and the OASIS exchange frequency for meteorological fields are set to the same value. Hence, CHIMERE may receive fields at a sufficient rate to avoid the need for a temporal interpolation of meteorological fields. The first version of the WRF-CHIMERE online coupling includes the exchange of the 28 WRF meteorological fields from WRF to CHIMERE through the OASIS coupler. Even though there is no feedback (i.e. exchange from CHIMERE to WRF) the use of instant WRF fields instead of interpolated fields will have an impact on the simulated results (see Section 5).

### 2.3.2 Aerosol optical properties feedback

The second version of the WRF-CHIMERE online coupling includes an aerosol optical properties feedback in order to take into account the direct and semi-direct aerosol effects. The feedback consists in 23 three-dimensional variables which are the single scattering albedos (SSA) and the asymmetry factors (AF) at 400nm and 600nm along with the AOD at 300nm, 400nm, 999nm and at 16 long-wavelengths ranging from 3400mn to 55600nm.

Short-wave aerosol optical properties are already calculated within CHIMERE using the Fast-JX model for radiative transfer and online calculation of photo-chemical rates (Wild et al., 2000; Bian and Prather, 2002). The computation of long-wave parameters is done following the same method, by extending the radiative properties calculations within CHIMERE to the required long-wavelengths.

Aerosol optical properties computed by CHIMERE are interpolated over WRF vertical grid before being sent through the OASIS coupler. If the CHIMERE top level is lower than the WRF top level, the optical properties climatology from Péré et al. (2014) is used for short-wave aerosol optical properties, for highest vertical levels. Long-wave aerosol optical properties of highest vertical levels are set to zero above CHIMERE top level.

Within the WRF model, short-wave AOD are interpolated over the required wavelengths using an Ångström power law, while the SSA and AF at 440nm and 600nm are interpolated assuming a linear relation. The long-wavelength AOD are added to the gases optical depth.



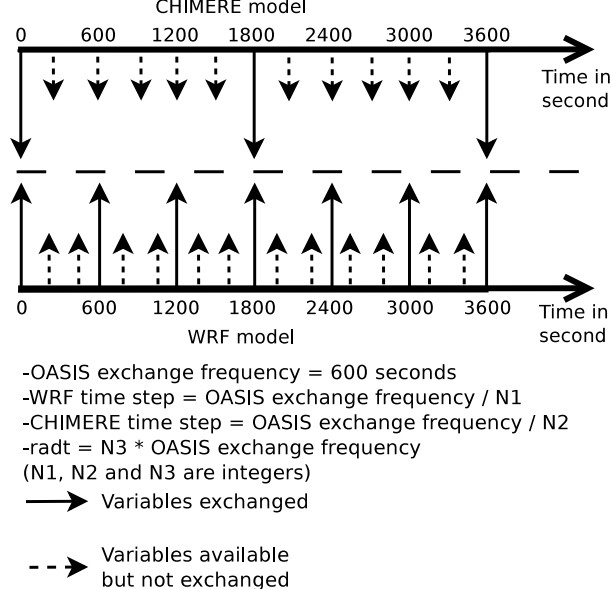

**Figure 1.** Illustration of variable exchange frequencies. The CHIMERE model receives WRF meteorological fields and sends the aerosol optical properties. The WRF model receives the aerosol optical properties and sends the meteorological fields. The OASIS exchange frequency defines both the frequency at which meteorological fields are exchanged (fixed here at 600 seconds) and the frequency at which aerosol optical properties are exchanged (fixed here at 1800 seconds, with N3 = 3). Both models may perform sub-iterations (here N1 = 3 and N2 = 2). Note that the OASIS exchange frequency along with the N1, N2 and N3 integers are parameters that may be set by users.

### 2.3.3 Exchanges from CHIMERE to WRF

Aerosol optical properties are sent from CHIMERE to WRF through the OASIS coupler and used within the WRF model as inputs for the RRTMG (Rapid Radiative Transfer Model for General circulation models) scheme (Iacono et al., 2008). WRF "radt" parameter sets the frequency at which the RRTMG scheme is called within the WRF model. The recommendation from the WRF user's guide is to set the "radt" parameter to 1 minute per kilometer of the grid distance between each grid cell (http://www2.mmm.ucar.edu/wrf/users/docs/user_guide_V3/ARWUsersGuideV3.pdf). As this frequency may be different

5   from the OASIS exchange frequency for meteorological fields, "radt" is fixed to a multiple of this OASIS exchange frequency. Therefore, whenever WRF requires the aerosol optical properties, CHIMERE is able to send it. Regardless of the exchange frequency value, both WRF and CHIMERE may perform sub-iterations to ensure that the Courant-Friedrichs-Lewy condition is satisfied (see Figure 1).

### 2.4 Operations scheduling

10   In case of a WRF-CHIMERE online coupled simulation without any feedback, OASIS exchanges are made in one way only (i.e. from WRF to CHIMERE). The operations scheduling is similar to what is done in offline mode, as CHIMERE is forced by



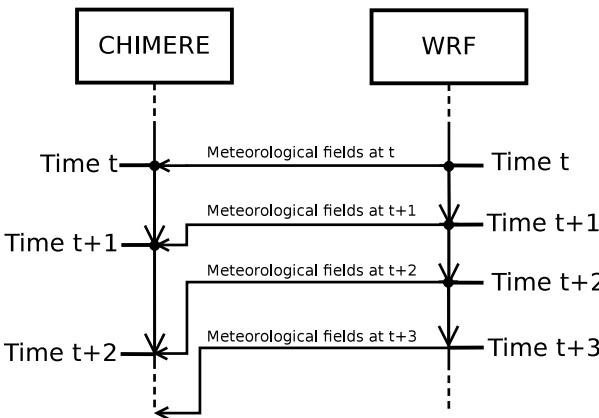

**Figure 2.** Operations scheduling in a WRF-CHIMERE online simulation with OASIS exchange from WRF to CHIMERE only.

WRF meteorological fields, but with a higher frequency. The initial meteorological fields sent are the WRF initial conditions. In Figure 2 the WRF model runs faster than the CHIMERE model, leading to accumulated delay between OASIS subsequent send and receive operations. However, as OASIS send operations are non-blocking, the WRF model may continue its calculations without having to wait for OASIS receive instructions within the CHIMERE model. In case of a CHIMERE model that would run faster than the WRF model, the CHIMERE model would wait for WRF meteorological fields.

In case of two-ways exchanges, the aerosol optical properties exchanges are performed right after the meteorological fields exchanges (i.e. at the beginning of each model time iteration). This allows the two models to perform their time iterations concurrently, thereby optimizing the overall computational burden (Figure 3). Initial aerosol optical properties that are sent to WRF may be provided as an input file in CHIMERE, if available, or are set to zero otherwise. When the aerosol optical properties feedback is activated, the two models may need to wait for each other in order to receive the required fields that will allow them to continue the run. In any case, the overall WRF-CHIMERE online simulation time is expected to be close to the maximum of both WRF and CHIMERE offline run times. Nonetheless, an increase of the computational time is expected within the CHIMERE model due to more frequent calls to meteorology treatment routines along with calls to optical properties computations routines.

## 3 Test case presentation

In order to evaluate both the computational burden and the model performances three simulation types are defined:

- Offline: both WRF and CHIMERE are run sequentially. CHIMERE reads meteorological fields at a hourly rate from WRF output file and the aerosol optical properties are not exchanged.

- Online case 1: WRF and CHIMERE are run online. Meteororological fields are sent through the OASIS coupler with a high temporal resolution (from WRF to CHIMERE). The aerosol optical properties feedback is not exchanged.



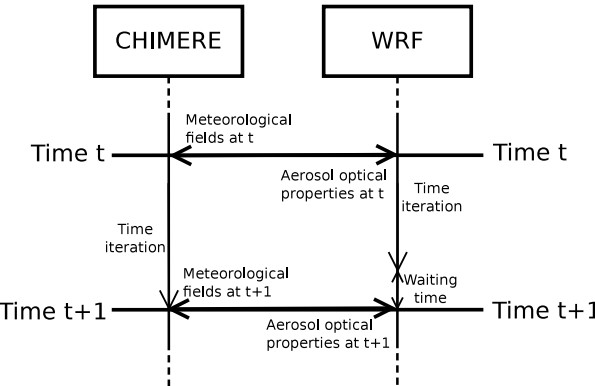

**Figure 3.** Operations scheduling in a WRF-CHIMERE simulation with the aerosol optical properties feedback.

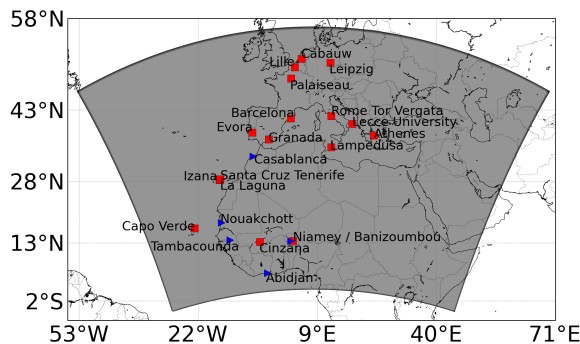

**Figure 4.** Simulated domain, used in Section 4 and 5. AERONET stations are located with red squares and temperature atmospheric sounding stations with blue triangles.

– Online case 2: WRF and CHIMERE are run online. Meteororological fields with a high temporal resolution (from WRF
    to CHIMERE) along with the aerosol optical properties (from CHIMERE to WRF) are sent through the OASIS coupler.

    from WRF to CHIMERE with a high temporal resolution. The aerosol optical properties are sent from CHIMERE to
    WRF.

The simulated domain horizontal grid was built with a Lambert projection and has $159 \times 109$ points in longitude and latitude.
It covers both Europe, northern Africa, Middle East and western Asia with a $60\,\mathrm{km}$ resolution (Figure 4).

Both offline and online simulations are run with the same configuration. Note that both the WRF and CHIMERE versions
that are used to perform all simulations presented in this paper are modified versions of the WRF 3.7.1 and CHIMERE2016a
releases. These versions may be ran in either offline or online mode and modifications from the releases include exclusively the





online modeling developments described in Section 2. Both WRF and CHIMERE configurations are presented in Section 3.1 and Section 3.2, respectively.

## 3.1 WRF model configuration

The WRF model is used in its non-hydrostatic configuration (Skamarock et al., 2007) and forced by the meteorological analysis data of NCEP/GFS (Kalnay et al., 1996) provided on a regular $1.125° \times 1.125°$ grid. The model is ran with 32 vertical levels, from the surface to $20\,\mathrm{hPa}$, and with a 150 seconds integration time step. The RRTMG scheme, mandatory for the aerosol

optical properties feedback, is used for both long and short wave radiations along with the Morrison 2-moment microphysics scheme (Morrison et al., 2009). The surface layer scheme is the MM5 similarity theory scheme (Beljaars, 1995) and the surface physics scheme is the unified Noah land-surface model (Tewari et al., 2004). The Mellor-Yamada Nakanishi Niino (MYNN) planeray boundary layer's surface layer scheme (Nakanishi and Niino, 2006, 2009) is used and the cumulus parameterization is based on the Grell-Freitas scheme (Grell and Freitas, 2014).

## 10 3.2 CHIMERE model configuration

The CHIMERE model takes into account four types of emission. Anthropic emission fluxes are pre-calculated fields from the HTAP 2010 inventory (Hemispheric Transport of Air Pollution), prepared by the EDGAR team http://edgar.jrc.ec.europa.eu/national_reported_data/htap.php. Both biogenic and mineral dust emission fluxes are computed within the CHIMERE model using the MEGAN emissions scheme (Guenther et al., 2006) for the biogenic emissions and the dust production model de-

scribed in Menut et al. (2015) for the mineral dust emissions. Finally, emissions related to biomass burning are pre-calculated using the model described in Turquety et al. (2014). The LMDZ-INCA global model climatology (Folberth et al., 2006) is used for aerosol and gases boundary conditions while the GOCART model is used for mineral dust boundary conditions (Ginoux et al., 2001). The MELCHIOR2 chemical mechanism and the Bessagnet et al. (2004) aerosol module are used. The Fast-JX module, version 7.0b (Wild et al., 2000; Bian and Prather, 2002), was included within the CHIMERE model in order to com-

pute photolysis rates along with aerosol optical depth (Mailler et al., 2016). Dry and wet depositions are treated as described in Wesely (1989) and Loosmore and Cederwall (2004). 20 pressure dependant vertical levels are used, from the surface up to $200\,\mathrm{hPa}$. The WRF model fields computed on 32 $\sigma$-levels, that are either received via the OASIS coupler (online mode) or read from the WRF output files (offline mode), are linearly interpolated over the 20 CHIMERE vertical levels.

Rea et al. (2015) studied the contribution of the different aerosol sources to surface particulate matter (PM), using the

CHIMERE model with a similar configuration and over a similar domain during the summer 2012. Results showed that both mineral dust and anthropogenic sources are the main contributors of PM over Europe and the Mediterranean region. Daily exceedances of the PM10 European Union limit ($50\mu\mathrm{g.m}^{-3}$) are captured at the right time. However the number of exceedances is generally overestimated by the model, particularly in the northern part of the domain.





## 4  WRF-CHIMERE computational performances

WRF and CHIMERE offline simulation times along with WRF-CHIMERE online simulation times are compared in this section. Tests consist in 24 hours simulations, that are ran on a 64 cores computer using the simulation domain and model configurations presented in Section 3. The exchange frequency, is set to 15 minutes for both ways exchanges, therefore a total of 96 exchange time steps is performed. Several test simulations are ran with different number of cores, which are equally distributed among WRF and CHIMERE models.

Considering the size of the domain and variable's dimensions, the total number of exchanged points per iteration is over 6.4 million for WRF to CHIMERE exchanges. When adding the aerosol optical properties feedback it lead to a total of 19.2 million of exchanged points per time iteration among the two models for both ways exchanges. An estimation of the computational burden of these variable exchanges is made in Section 4.1, calculation and waiting times are studied using the LUCIA utility, that is distributed together with OASIS (Maisonnave and Caubel, 2014), in Section 4.2 and the load balance of each model is discussed in Section 4.3.

### 4.1  Comparison of both offline and online simulation times

The total online simulation duration are compared here to the offline simulation times. Time measurements were made using the Linux command: "time". There is an uncertainty regarding these measurements that is not fully known as it depends on the load of the computer that is used, which may vary during the simulations. However, simulations are long enough for the average times per iteration along with the trend to be significant.

Average simulation times per iteration are shown in Figure 5 as a function of the number of cores per model. As the WRF model is much faster than the CHIMERE model, the maximum of both WRF and CHIMERE offline run times is equal to CHIMERE offline run time. As expected, the CHIMERE model parallelization induces a decrease of the overall computational time with the increase of the number of cores. The decrease tendency is preserved in both online simulations, however both online cases require more computational resources. The time increase is higher for online case 2 simulation than for online case 1 simulation, as more variables are exchanged and more computations are made (see Section 2.3). Highest time increase occurs when a lower number of cores is used (up to 170 seconds increase using 1 core per model in the case 2 simulation). On the other hand, the percentage of time increase from the offline simulation increases with the number of cores and reaches a 42% increase when using 32 cores per model in the case 2 simulation (Figure 6). Part of the additional burden may be attributed to the OASIS exchange along with the variable formatting routines. The other part is related to additional calls to some CHIMERE routines that are made in online mode (i.e. more frequent meteorology treatment subroutine in case 1 along with optical properties computations in case 2). A measure of the computational burden that may be attributed to variable exchange subroutines has been made using the Fortran routine "cpu_time". These subroutines are responsible for less then 3% of the time increase, for both online cases when using 32 cores per model. Therefore the increase of computational burden that may be attributed to the OASIS exchange is not significant compared to the model computations.



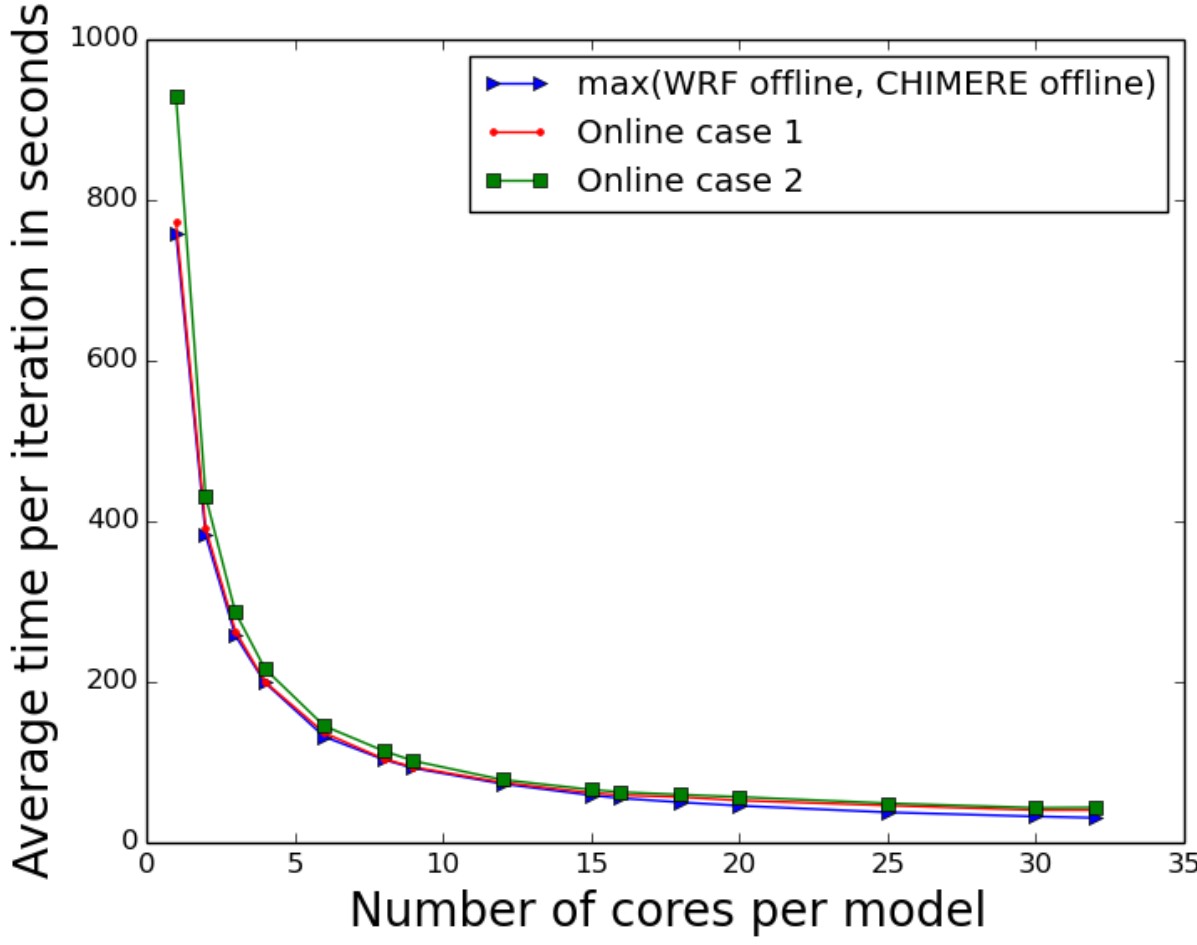

**Figure 5.** Evolution of the computational time per iteration as a function of the number of core per model. Online case 1 refers to the online simulation without the aerosol optical properties feedback and online case 2 refers to the online simulation with the aerosol optical properties feedback.





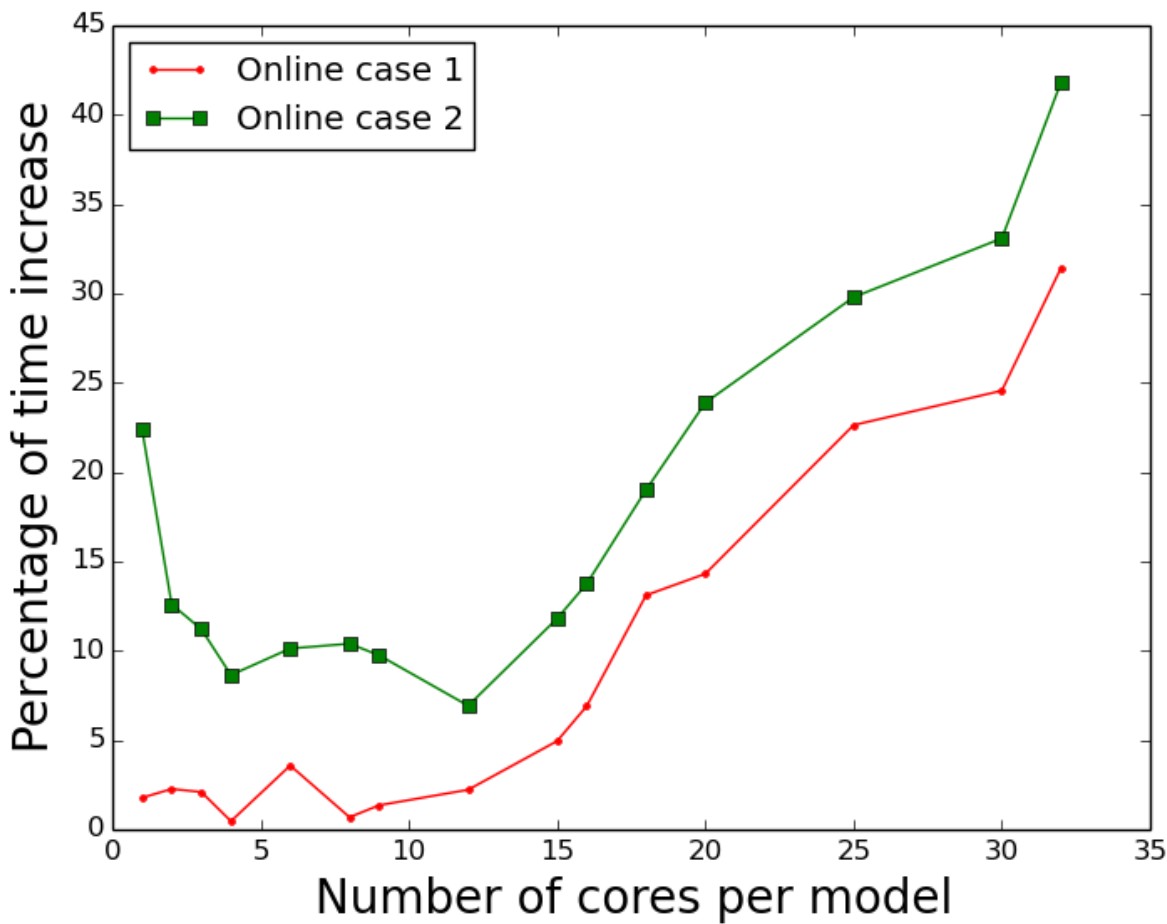

**Figure 6.** Evolution of the time increase from the offline simulation of both online simulations, as a function of the number of core per model. Online case 1 refers to the online simulation without the aerosol optical properties feedback and online case 2 refers to the online simulation with the aerosol optical properties feedback.





## 4.2 Calculation and waiting times

Results presented in this section were obtain using the LUCIA utility on the 32 cores per model simulations, which provide the total calculation and waiting times for of both models and for both case 1 and case 2 simulations. Online case 1 results indicates that WRF performs less calculations than CHIMERE, i.e 770 seconds for WRF vs 3630 seconds for CHIMERE. This is consistent with the fact that there is almost no waiting time for the CHIMERE model (i.e. 10 seconds), as WRF

meteorological fields are always available when CHIMERE required them. Even though OASIS send operations are non-blocking, the WRF model waits for CHIMERE during 2890 seconds. A possible explanation is that WRF is so much ahead of CHIMERE that its sending buffer is full. Thus, WRF needs to wait for CHIMERE receive instructions to empty its buffer and to continue the run. Nevertheless, as CHIMERE is computationally more costly, WRF waiting times do not induces any additional burden to the overall WRF-CHIMERE online simulation. Similar results are observed for the case 2 simulation. As

both model iterations are done in parallel, the aerosol optical properties feedback do not induce a significant change in the overall balance between the two models.

## 4.3 Load balance of each model

In both online cases the WRF model performs less calculations than the CHIMERE model. Attributing the same number of cores to both models was an arbitrary choice that needs to be revised. Attributing a lower number of cores to WRF and a higher

number of cores to CHIMERE shall reduce the overall computational time. Section 4.2 results indicate that using 4 or 5 times more cores with CHIMERE than with WRF may be an efficient ratio. As the unbalance load of each model may also depend on other criteria such as the selected options within both WRF and CHIMERE configuration files or the geometry of the domain, the ratio of cores that will optimize the computational burden is not unique. Iterative methods may be employed to estimate the optimum ratio of number of cores for each model.

# 5   WRF-CHIMERE evaluation study

WRF-CHIMERE online simulations are confronted in this section to both measurements and a corresponding offline simulation. The simulated period starts on the $15^{th}$ of May 2012 and ends on the $14^{th}$ of July 2012, therefore, covering the June 2012 mineral dust outbreak event (Nabat et al., 2015). Simulated results from $15^{th}$ until $31^{st}$ of May are considered as spin-up time. Thus only the simulated results from the $1^{st}$ of June are considered for the evaluations made in the following sections. The

OASIS exchange frequency for meteorological fields, thus the CHIMERE physical time step is set to 15 minutes and WRF "radt" parameter is set to 30 minutes. WRF meteorological fields and CHIMERE output concentrations are stored every hour for the analysis.

Simulated radiation budget, surface temperatures and wind velocities are compared in Section 5.1. Simulated results are then successively evaluated against University of Wyoming vertical temperature atmospheric soundings (Section 5.2), MODIS

AOD (Section 5.3), AERONET AOD (Section 5.4) and AirBase PM10 concentrations data (Section 5.5).



### 5.1 Feedback impact on radiation budget, surface temperatures and wind velocities

The radiative forcing is defined as the difference in the net radiation flux (down-up) between both online simulations. Changes in the radiation budget induced by the optical properties feedback is studied here through the radiative forcing induced by the aerosol optical properties feedback. Figure 7 shows difference maps between the two online cases of the average radiation budget at the ground surface for both long-waves and short-waves. Long-wave radiative forcing attributed to the optical properties feedback in case 2 simulation is positive, up to $35\mathrm{W.m^{-2}}$, and is mainly located over desert areas (i.e. Saharan region and the Arabian peninsula). A negative forcing is observed over the Atlantic ocean of a lesser importance, less than $5\mathrm{W.m^{-2}}$. An opposite behavior is obtained for short-wave radiation fluxes, as there is a negative forcing, up to $55\mathrm{W.m^{-2}}$ over the Saharan region and the Arabian peninsula and a positive forcing of a lesser importance over the Atlantic ocean, less than $28\mathrm{W.m^{-2}}$. The average forcing over the simulated domain is a cooling of $4.8\mathrm{W.m^{-2}}$ (i.e. radiative forcing of $5.8\mathrm{W.m^{-2}}$ for long-waves and $-10.7\mathrm{W.m^{-2}}$ for short-waves).

The perturbation of the WRF radiative scheme outputs depends on the CHIMERE aerosol optical properties, thus on the CHIMERE aerosol load. In our case the perturbation in the optical properties is dominated by mineral dust, as observed changes occur over region where mineral dust constitute the main aerosol type (i.e. Saharan region and Arabian peninsula). Mineral dust both absorbs and scatters solar radiation, leading to both negative and positive radiative forcing depending on the radiation wavelength and on the mineral dust size distribution (Sokolik and Toon, 1996). Aerosol absorption of solar radiation induces a heating of the atmosphere, thus a reduction of the cloud coverage. This effect is referred to as the aerosol semi-direct effect (Hansen et al., 1997; Ramanathan et al., 2001) and is responsible for part of the changes in the radiative forcing. Off the western Saharan coast, high mineral dust concentrations cause a reduction of the cloud coverage, therefore inducing an increase of the short-wave radiative forcing in online case 2 simulation.

In Guo and Yin (2015) the mineral dust impacts on the regional precipitation and summer circulation in East Asia are studied. A negative short-wave radiative forcing along with a positive long-wave radiative forcing induced by the presence of mineral dust particles are observed. The long-wave radiative forcing is less than $50\mathrm{W.m^{-2}}$ and the short-wave radiative forcing is lower than $-70\mathrm{W.m^{-2}}$. Even though the simulated areas are different, the impacts of mineral dust on the radiative forcing are in accordance with the results presented in the current paper.

A direct consequence of the changes in the radiative forcing is a perturbation of the surface temperatures. Figure 8 maps show a moderate decrease of the surface temperatures (i.e. less than $0.4°$) over the Sub-Saharan Africa, Europe and over the northern part of the Atlantic ocean. Over the Saharan region, the Arabian peninsula and off the western Saharan coast temperatures increase up to a local maximum of $2.6°$ over a grid cell in north-east Niger.

Figure 9a presents a four-days time serie (1[st] to 4[th] of June) of surface temperatures over the north east Niger grid cell in which the maximum differences of average surface temperatures occurs. The diurnal profile shows that an increase of temperatures occurs during nighttime (up to $5°$) while a slight decrease of temperatures occurs during daytime (less than $1°$). Figure 9b shows that the short-wave effect prevails during daytime, thus creating a decrease of the surface temperatures, while





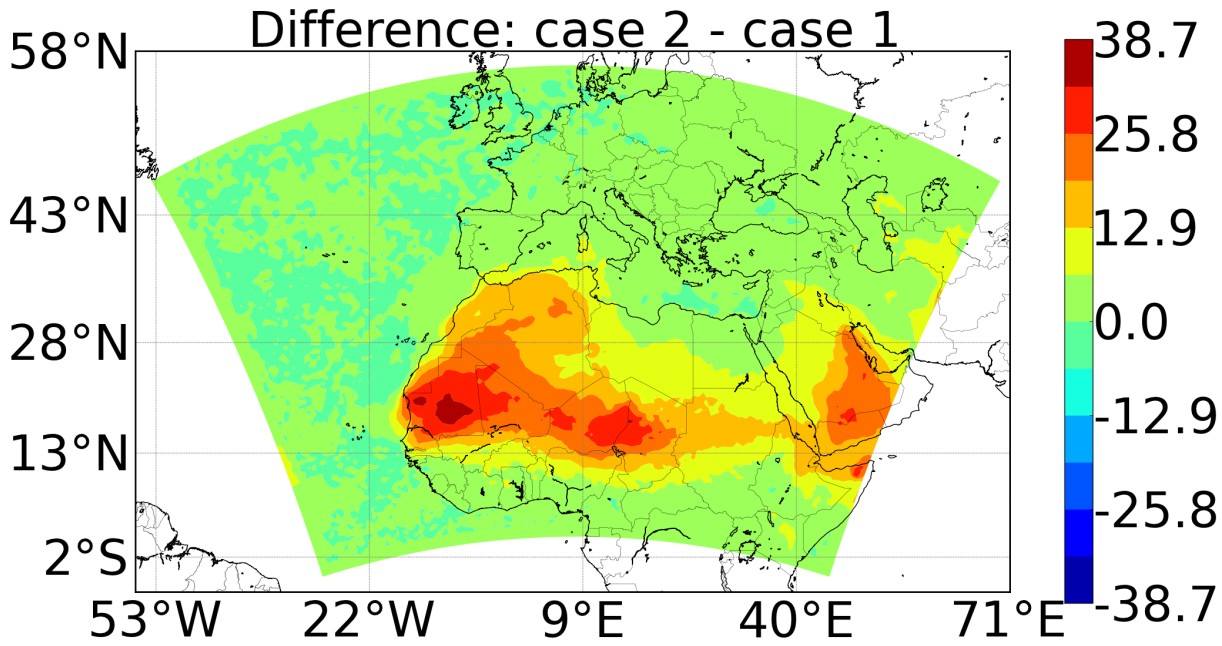

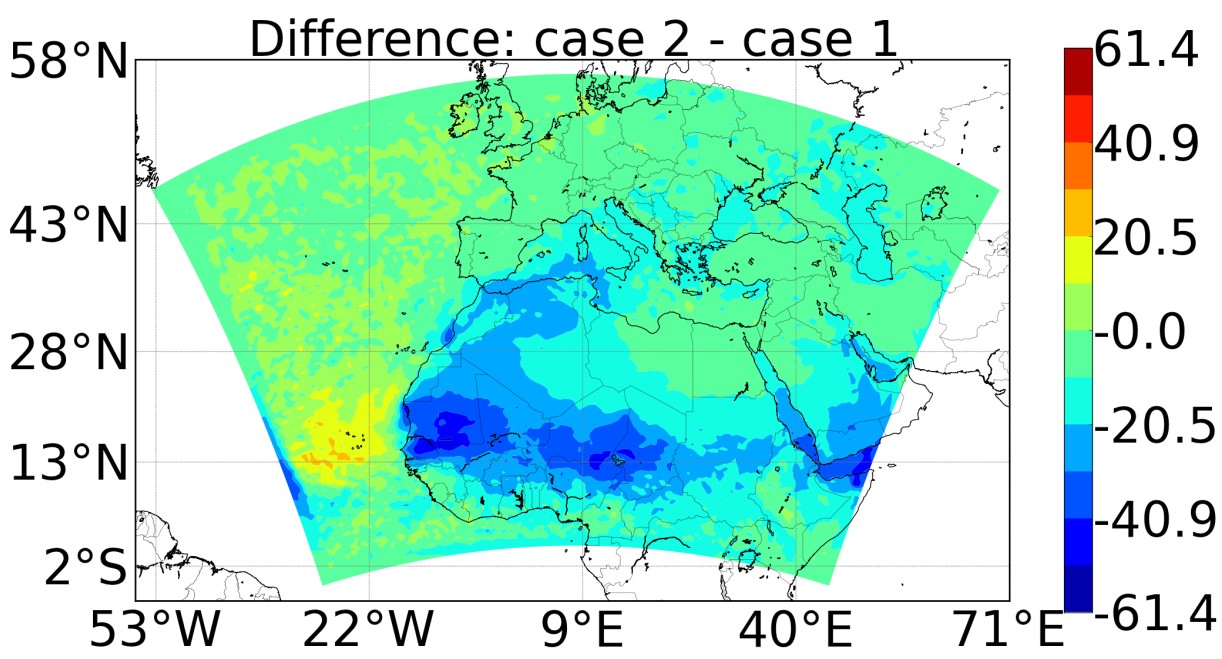

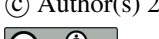



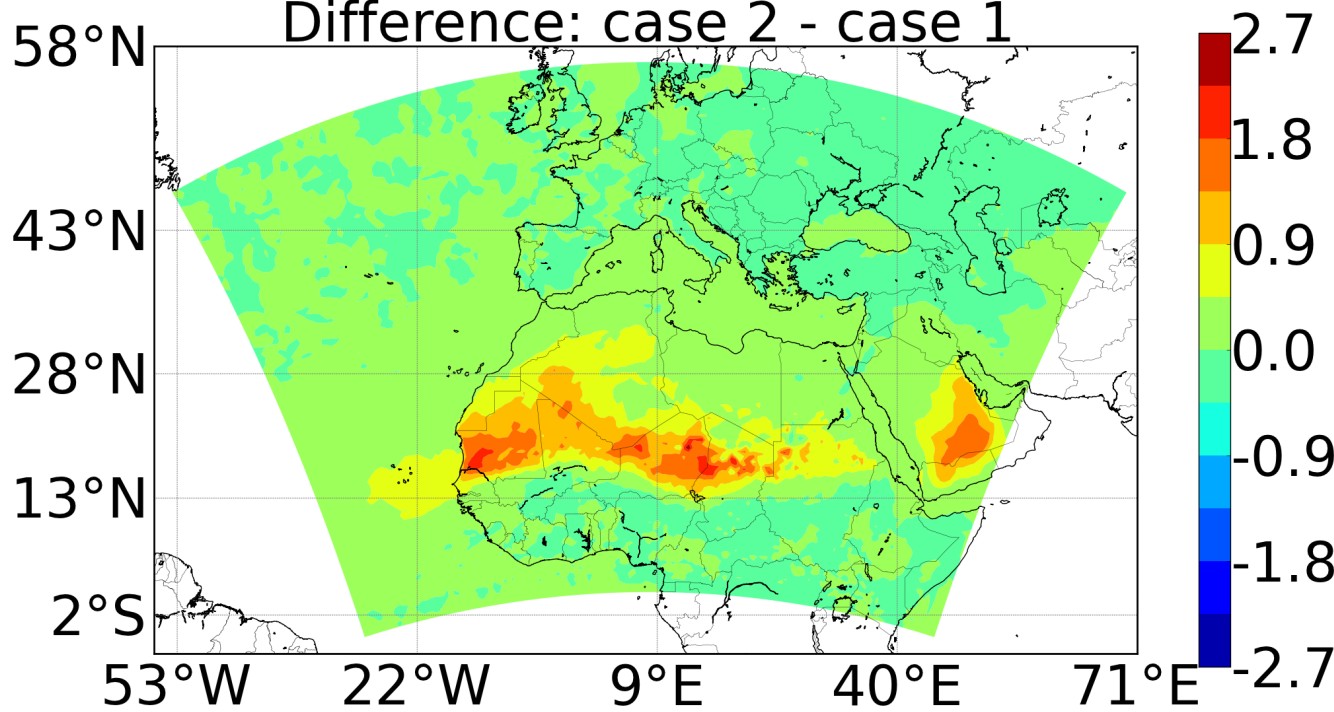

**Figure 8.** Difference map of WRF temperature at 2 meters from the surface averaged over the simulated period ranging from the $1^{st}$ of June to the $14^{th}$ of July (in Kelvin).

the long-wave effect alone contributes at night due to the earth outgoing long-wave radiations, inducing an increase of the temperatures. This is also observed in Yue et al. (2010); Guo and Yin (2015).

Another consequence of the perturbation of the radiative forcing is the alteration of the wind velocities. Figure 10 shows that the use of the aerosol optical properties feedback in online case 2 simulation induces both an increase (up to $0.5 \mathrm{~m.s}^{-1}$) and a decrease (up to $0.4 \mathrm{~m.s}^{-1}$) of the wind module over part of the Saharan region and the Arabian peninsula. As the wind velocity is the main parameter influencing mineral dust emissions, changes in CHIMERE aerosol content is also expected.

5 ## 5.2 Comparison with the University of Wyoming atmospheric sounding vertical temperature data

Atmospheric sounding temperature data were gather at 5 stations over the Saharan region and the Arabian peninsula (see Figure 4 for station locations), from the University of Wyoming website (http://weather.uwyo.edu/upperair/sounding.html). Differences of temperature vertical profile between sounding and online modeled values are displayed in Figure 11 at selected times. Results are interpolated over the soundings vertical levels. Stations are located in western Sahara (Tambacounda, Abid-
10 jan, Nouakchott and Niamey) where the impact of mineral dust emissions, thus differences in solar radiation, is important. Profile are shown for the $23^{rd}$ of June, during the end of June mineral dust outbreak (i.e. from $21^{st}$ to $23^{rd}$ of June). In addition, temperature vertical profiles are shown at the Casablanca station at both the $23^{rd}$ and $26^{th}$ of June. Therefore, the two







(a) Surface temperature (in Kelvin)

(b) Downwelling radiative forcing: case 2 - case 1 (in W.m$^{-2}$)

**Figure 9.** Surface temperature and downwelling radiative forcing four-days time series (1$^{st}$ to 4$^{th}$ of June) over a grid cell in north-east Niger.





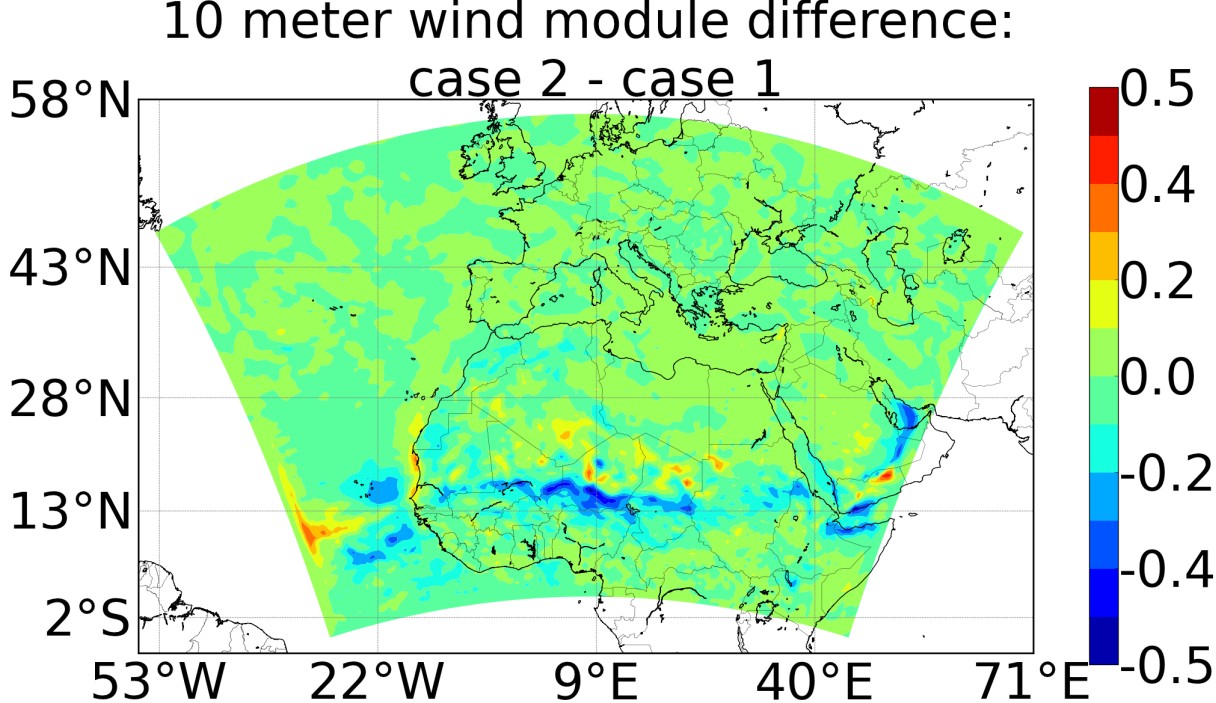

**Figure 10.** 10 meter high wind module difference map between online case 1 and case 2 (in $\mathrm{m.s}^{-1}$) averaged in time over the period ranging from the $1^{\mathrm{st}}$ of June to the $14^{\mathrm{th}}$ of July.

profiles at the Casablanca station allow to compare vertical temperature profiles with a low and with a high level of mineral dust.

15     Modeled values tend to overestimate the observations at higher levels and underestimate the observations at some lower levels. Differences between observations and modeled values lie between -6.5° at the Nouakchott station and 3.1° at the Tambacounda station. Differences among modeled values are not significant at higher levels (roughly above 5000 meter, where mineral dust concentrations are low) and is less than 0.12° at the highest level at all stations, except for the Casablanca station on the $26^{\mathrm{th}}$ of June where the temperature difference between both online cases at the highest level is 0.5°. Therefore, only the

5   lower part of vertical profiles is shown on Figure 11. At the Tambacounda station the online case 2 simulation performs better below 1000 meter. The online case 2 simulation reduces the underestimation of measurements by 0.6°. Above 1000 meter each of the two simulations perform better alternatively. At the Abidjan station the online case 2 simulation has lower values which





reduce the overestimation of measurements over the whole profile. On the contrary, at the Nouakchott station the online case 2 simulation has higher values than the online case 1 simulation, inducing an increase of the overestimation of measurements (up to 0.7° higher). At the Niamey station the online case 2 simulation performs better by reducing the temperature by up to 0.5° at the lowest level, therefore reducing the overestimation of measurements. However, the Niamey profiles has 5 vertical levels only. On the 23$^{rd}$ of June at the Casablanca station difference among modeled values are low (less than 0.3°) compared to the 26$^{th}$ of June differences (up to 1.7°). The end of June mineral dust event increases mineral dust concentrations, thus increases

the AOD that is transmitted to WRF RRTMG scheme in the online case 2 simulation.

Online case 2 simulation may result in both higher or lower temperatures and induce either improved or diminished performances. However, the impact of the aerosol optical properties feedback is important with differences up to 1.7°.

## 5.3    Comparison with MODIS AOD

The Moderate Resolution Imaging Spectroradiometer (MODIS) satellite data are compared to the modeled AOD (Levy et al.,

2015). MODIS Dark-Target and Deep-Blue products at 550nm are merged in order to form a single map. Deep-Blue product is preferred when both products are available at a given point, as it is more accurate over desert areas (Hsu et al., 2013). Data are averaged over the period ranging from the 1$^{st}$ of June to the 14$^{th}$ of July. Modeled values were also averaged in time, using only modeled values at times at which a MODIS observation is available. As CHIMERE aerosol optical depth is calculated at fixed wavelengths (i.e. 200nm, 300nm, 400nm, 600nm and 999nm), the AOD is interpolated at 550nm following

an Ångström power law. The corresponding MODIS AOD map is displayed on the top left corner of the Figure 12 and the difference between MODIS and offline AOD is shown on the top right corner of the Figure 12. In addition, both online AOD are shown as difference among modeled values rather than difference with the MODIS AOD (bottom of Figure 12).

Over major sources of mineral dust, such as the Saharan region and the Arabian peninsula, the MODIS AOD values are high (up to 1.3). However, even higher values are observed over the eastern side of the Caspian Sea, the Red Sea and the

Zagros mountain (up to 3.). Both offline and online simulations failed to detect these high values over those three regions. Such CHIMERE / MODIS AOD differences were already observed in Menut et al. (2015) and in Mailler et al. (2016). Over the eastern part of the Caspian Sea, those differences may be attributed to missing mineral dust as it is an arid region. In Nabat et al. (2015) the MODIS data overestimates the satellite product MISR (Multiangle Imaging SpectroRadiometer (Kahn et al., 2005)) and AERUS-GEO (Aerosol and surface albedo Retrieval Using a directional Splitting method; application to GEO data

by Carrer et al. (2014)) over the Red Sea and on the eastern side of the Caspian Sea. This suggests that the high MODIS AOD values may be attributed to an overestimation made by the MODIS aerosol retrieval algorithm.

Over Europe, North Africa and the Atlantic ocean, differences between MODIS and the offline simulated AOD are less than 0.4. Major differences occur on the western part of the Sahara (south of Mauritania) and on the southern part of the Arabian peninsula, where the CHIMERE model overestimates the MODIS AOD by up to 1.4. Differences are most likely due to an

overestimation of mineral dust emissions, which are the main AOD contributor in those areas.

The more resolved meteorology in online case 1 simulation mainly induces higher AOD than the offline simulation. The AOD increase is ranging from 0.03 over Europe up to 0.08 over south Mauritania and western Mali. Changes induced by the




aerosol optical properties feedback are more important (difference up to 0.25), however it induces both increases and decreases, principally over both Africa and the Arabian peninsula. Differences may be explained by the alteration of the wind velocities, thus of the mineral dust emissions, in these areas (Figure 10).

## 5.4 Comparison with AERONET AOD

Daily AOD at 675nm of both level 2.0. quality assured AERONET data (Holben et al., 1998) and CHIMERE AOD, are compared in this section. The location of AERONET stations is shown in Figure 4.

A mineral dust outbreak occurred over Western Africa between the $21^{st}$ and $23^{rd}$ June (Nabat et al., 2015). Due to a lack of data during this period, this mineral dust outbreak is not visible on the AOD time series at the Capo Verde and Cinzana stations. However, particles have been transported along the African coast up to southern Spain, therefore it is visible on the AOD time series at the Izana, La Laguna, Santa Cruz Tenerife ($24^{th}$ to $30^{th}$ of June), Granada ($24^{th}$ to $30^{th}$ of June), Evora ($25^{th}$ to $29^{th}$ of June) and Barcelona ($27^{th}$ of June to $1^{st}$ of July) stations (Figure 13). Even though, AOD peak intensities tend to be

overestimated, models manage to predict efficiently the times at which high AOD events occur. Although, high AOD events are detected at the same moment in each simulation, variations in the peak intensities appear. However, time series alone are not sufficient to infer whether or not one simulation performs better than another because the three simulation results are close to each other.

    AOD performance indicators over the period ranging from the $1^{st}$ of June to the $14^{th}$ of July are shown in Table 1 and are

defined as:

- Correlation:

$$\frac{\sum_{i=1}^{N}(O_i - \overline{O})(M_i - \overline{M})}{\sqrt{\sum_{i=1}^{N}(O_i - \overline{O})^2}\sqrt{\sum_{i=1}^{N}(M_i - \overline{M})^2}}$$

- RMSE (root mean square error):

$$\sqrt{\frac{1}{N}\sum_{i=1}^{N}(M_i - O_i)^2}$$

- Bias:

$$\frac{1}{N}\sum_{i=1}^{N}(M_i - O_i)$$

where $M_i$ and $O_i$ are the modeled and observed values, respectively and $\overline{x} = \frac{1}{N}\sum_{i=1}^{N}x_i$.

    Apart from the Lampedusa station, RMSE are less than 0.19 at all European stations while at African stations it ranges from 0.18 in La Laguna up to 0.55 in Capo Verde. Six stations have a particularly low correlation (less than 0.5), Cabauw (0.19

to 0.2), Cinzana (0.23 to 0.29), Banizoumbou (0.37 to 0.44), Lille (0.48), Capo Verde (0.47 to 0.51) and Leipzig (0.5) while correlations at other stations are higher, ranging from 0.67 at Palaiseau up to 0.95 at Evora and Izana. Bias are higher at both





the Izana, Capo Verde and Lampedusa stations (from 0.14 to 0.38) and is less than 0.08 elsewhere. The three African stations located near major mineral dust sources (i.e. Banizoumbou, Cinzana and Capo Verde) present lower performances. This may be explained by the difficulty to reproduce mineral dust event within the model, as mineral dust is the main AOD contributor at these stations. If a mineral dust event is not detected or if it is wrongly detected by the model, the impact on the AOD may be important.

Models overestimate measurements at 12 out of 17 stations. Furthermore, average AOD are higher in both online simulations
than with the offline simulation. The offline simulation perform equivalently or better at European stations (higher correlation and lower RMSE and bias), however simulated results are close to each other. Differences among modeled values are higher at African stations (mean value differences up to 0.6) than at European stations (mean value differences up to 0.2 at the Barcelona station). The online case 2 have higher correlations (up to 0.4 higher) and lower RMSE (up to 0.2 lower) at the Izana, Santa Cruz Tenerife, Capo Verde and La Laguna stations than the other simulations. At both the Banizoumbou and Cinzana stations
the offline simulation presents higher correlations and lower negative biases than the online simulations.

## 5.5 Comparison with AirBase PM10 concentrations

Hourly PM10 measurement from the European Air quality database (AirBase) of the European Environment Agency (http://acm.eionet.europa.eu/databases/airbase) are used in this section for comparison with CHIMERE PM10 concentrations. As in Rea et al. (2015), only rural and background stations are considered for the comparison in order to avoid sites which are
strongly influenced by local sources. In addition, stations with a minimum of 300 measurements during the period ranging from the $1^{st}$ of June to the $14^{th}$ of July are selected, leading to a total of more than 940 remaining stations located over Europe.

Averaged performance indicators show that all simulations overestimate measurements and that the overestimation is higher with both online simulations ($6.8\mu g.m^{-3}$) than with the offline simulation ($1.7\mu g.m^{-3}$). Correlations are lower (differences up to 0.17) and RMSE are higher (differences up to $22\mu g.m^{-3}$) at most stations for both online simulations. The increase of PM10
concentrations in online simulations is consistent with results of Sections 5.3 and 5.4, in which the more resolved meteorology in online case 1 simulation induces higher AOD over Europe. Indeed, higher-frequency meteorological fields, received by CHIMERE from WRF, in the online simulations are associated with higher temporal variability, which are smoothed out through the temporal interpolation in the offline simulation. In case of the wind velocity for instance it can lead to higher mineral dust emissions, which is a threshold process, and/or particulate matter resuspension, thus increasing the PM10 concentrations
in online mode. A deeper analysis is needed, using PM10 concentration measurements over Africa, in order to assess the overall impact of the WRF-CHIMERE coupling on PM10 concentrations.

## 6 Conclusions

An online coupling between WRF and CHIMERE models through the OASIS coupler has been developed. WRF meteorological fields along with CHIMERE aerosol optical properties are exchanged in order to simulate the aerosol direct and semi-direct
effects.





The WRF-CHIMERE online model requires more computational resources than the offline models, mainly due to the CHIMERE model as the WRF model is less demanding. The computational time increase within the online model is mostly related to additional calls to some routines that are made in order to prepare the fields before being sent or to treat the received fields. On the other hand the increase of computational time related to OASIS exchanges is not significant. Therefore, increasing the amount of OASIS exchange in future development shall not be an issue.

Both offline and online simulations of two months of the summer 2012 are compared. The use of the optical properties feedback induces a $5.8\,\mathrm{W.m^{-2}}$ average increase of long-wave radiative forcing and a $10.7\,\mathrm{W.m^{-2}}$ decrease of short-wave

radiative forcing. Consequences of the radiative forcing perturbation are changes in the averaged surface temperatures (i.e. increase up to $2.6°$ over desert areas and a moderate decrease of less then $0.4°$ elsewhere) and wind velocities (i.e. averaged differences ranging from $-0.4\,\mathrm{m.s^{-1}}$ to $0.5\mathrm{m.s^{-1}}$). Diurnal profiles over the grid cell where the average temperature difference is maximum show that temperatures decrease slightly during daytime, when the short-wave effect prevails. On the other hand temperatures increase at night, when the long-wave effect alone contributes due to the earth outgoing long-wave radiations.

Therefore, the modeling of the aerosol direct and semi-direct effects, through the aerosol optical properties feedback, is not negligible. However, it may result in either improved or diminished performance, depending on the location. Observed AOD by the AERONET network are compared to modeled AOD, leading to higher correlations and lower RMSE at African stations when using the aerosol optical properties feedback. Over Europe, differences among simulations are less important than over Africa, where mineral dust emissions represent a more important part of the total aerosol content. Modeled PM10 concentra-

tions are higher, therefore increasing the overestimation of the AirBase PM10 concentrations. As a results AOD are higher and performances of AERONET AOD compared to modeled AOD are reduced.

Even though the radiative coupling between WRF and CHIMERE does not necessarily improve the model performances in terms of biases and correlations to the observations, these results open the way to the use of the WRF-CHIMERE coupled system to simulate cases where the radiative effects of optically thick aerosol plumes on the atmosphere is significant, and to

examine the impact of these dense plumes on meteorology, and their feedbacks on the advected plumes themselves. Results depicted in this paper emphasize that using the aerosol optical properties feedback induces non-negligible changes in model outputs. In addition, up to now emissions were designed for offline models and some modifications within emission parameterizations, mineral dust in particular, may be required in online mode. For instance, the more resolved meteorology in online simulation induces an increase of the wind velocity variability. However, a Weibull distribution is used to account for the wind variability within the mineral dust emission parameterization (Cakmur et al., 2004).

Online modeling developments presented in this paper will be made publicly available through a future CHIMERE release. The development of the WRF-CHIMERE online coupling continue with the implementation of another WRF-CHIMERE feedback, aiming at modeling the aerosol-cloud microphysical interactions. In addition, as the CHIMERE model is now interfaced

with the OASIS coupler, future work may involved online coupling with other models.



## 7  Code availability

The CHIMERE model is provided under the GNU General Public License and is available at http://www.lmd.polytechnique.fr/chimere/. Online coupling developments will be made available in a future CHIMERE release and are available upon request.

*Acknowledgements.* The authors acknowledge the french Direction Générale de l'Armement for funding the APPAD project, the COPERNICUS-DD program for funding the NATORGA project and the University of Geneva for its support. We thank the OASIS modeling team for their support with the OASIS coupler. Finally, we thank the National Aeronautics and Space Agency for the availability of the MODIS data, the investigators and staff who maintain and provide the AERONET data, the University of Wyoming for the availability of the atmospheric soundings, along with the European Environment Agency for providing the AirBase data.





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





**Figure 11.** Difference of vertical profile of temperature (radiosounding values - modeled values).





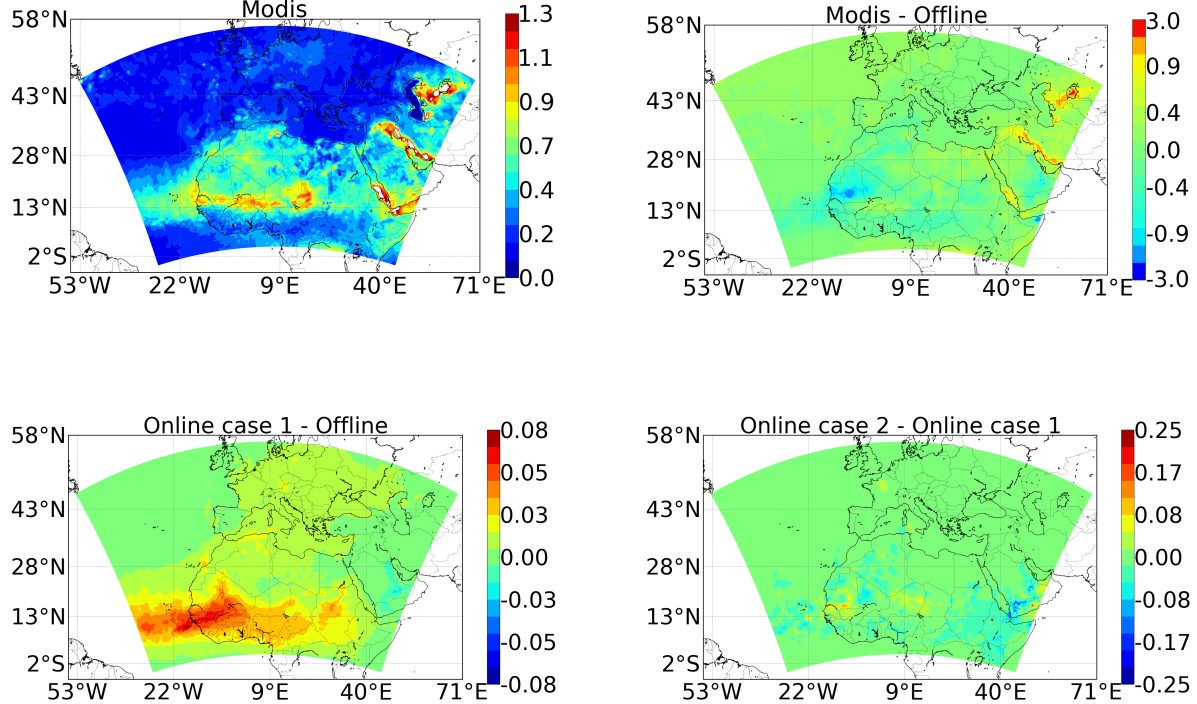

**Figure 12.** AOD and AOD differences maps at 550nm, averaged in time over the period ranging from the $1^{st}$ of June to the $14^{th}$ of July.





**Figure 13.** AERONET and modeled AOD time series.



**Table 1.** Performance indicators of WRF-CHIMERE modeled values against daily AERONET AOD measurements over the period ranging from the $1^{st}$ of June to the $14^{th}$ of July. Meas, Off, On1 and On2 correspond to measurements, offline simulation, online case 1 simulation and online case 2 simulation respectively. N is the number of observation and RMSE is the root mean square error.

| Station names | N | Mean values | | | | RMSE | | | Correlation | | | Bias | | |
|---|---|---|---|---|---|---|---|---|---|---|---|---|---|---|
| | | Meas | Off | On1 | On2 | Off | On1 | On2 | Off | On1 | On2 | Off | On1 | On2 |
| Izana | 44 | 0.12 | 0.28 | 0.29 | 0.29 | 0.25 | 0.26 | 0.24 | 0.94 | 0.94 | 0.95 | 0.16 | 0.17 | 0.16 |
| Lampedusa | 44 | 0.18 | 0.32 | 0.33 | 0.33 | 0.23 | 0.25 | 0.25 | 0.77 | 0.77 | 0.75 | 0.14 | 0.15 | 0.15 |
| Granada | 44 | 0.18 | 0.24 | 0.24 | 0.25 | 0.14 | 0.14 | 0.15 | 0.85 | 0.85 | 0.84 | 0.05 | 0.06 | 0.06 |
| Lecce University | 44 | 0.12 | 0.15 | 0.16 | 0.16 | 0.08 | 0.08 | 0.09 | 0.72 | 0.72 | 0.71 | 0.03 | 0.04 | 0.04 |
| Santa Cruz Tenerife | 42 | 0.26 | 0.3 | 0.31 | 0.3 | 0.21 | 0.21 | 0.2 | 0.86 | 0.87 | 0.89 | 0.03 | 0.04 | 0.04 |
| Evora | 42 | 0.11 | 0.12 | 0.13 | 0.13 | 0.09 | 0.1 | 0.1 | 0.95 | 0.95 | 0.94 | 0.02 | 0.02 | 0.02 |
| Rome Tor Vergata | 41 | 0.13 | 0.18 | 0.19 | 0.19 | 0.14 | 0.14 | 0.14 | 0.82 | 0.82 | 0.82 | 0.05 | 0.06 | 0.06 |
| Banizoumbou | 37 | 0.69 | 0.65 | 0.69 | 0.68 | 0.29 | 0.29 | 0.3 | 0.44 | 0.41 | 0.37 | -0.04 | 0.0 | -0.01 |
| Cinzana | 35 | 0.7 | 0.62 | 0.68 | 0.68 | 0.45 | 0.45 | 0.45 | 0.29 | 0.25 | 0.23 | -0.08 | -0.02 | -0.02 |
| Capo Verde | 31 | 0.53 | 0.87 | 0.91 | 0.88 | 0.53 | 0.55 | 0.51 | 0.47 | 0.48 | 0.51 | 0.35 | 0.38 | 0.36 |
| La Laguna | 31 | 0.29 | 0.34 | 0.35 | 0.34 | 0.2 | 0.2 | 0.18 | 0.89 | 0.89 | 0.91 | 0.05 | 0.06 | 0.06 |
| Athenes | 30 | 0.11 | 0.14 | 0.14 | 0.14 | 0.05 | 0.06 | 0.06 | 0.74 | 0.74 | 0.72 | 0.02 | 0.03 | 0.03 |
| Leipzig | 27 | 0.11 | 0.11 | 0.13 | 0.12 | 0.11 | 0.12 | 0.11 | 0.5 | 0.5 | 0.5 | 0.0 | 0.01 | 0.01 |
| Cabauw | 23 | 0.12 | 0.06 | 0.06 | 0.06 | 0.09 | 0.08 | 0.08 | 0.2 | 0.19 | 0.19 | -0.06 | -0.06 | -0.06 |
| Palaiseau | 22 | 0.11 | 0.07 | 0.08 | 0.08 | 0.08 | 0.07 | 0.08 | 0.67 | 0.67 | 0.67 | -0.04 | -0.03 | -0.03 |
| Lille | 21 | 0.1 | 0.07 | 0.07 | 0.07 | 0.09 | 0.09 | 0.09 | 0.48 | 0.48 | 0.48 | -0.03 | -0.03 | -0.03 |
| Barcelona | 20 | 0.2 | 0.23 | 0.25 | 0.24 | 0.17 | 0.18 | 0.19 | 0.8 | 0.8 | 0.77 | 0.03 | 0.04 | 0.04 |