# Peer review of "Aerosol-radiation interactions modeling using an online coupling between the meteorological model WRF 3.7.1 and the chemistry-transport model CHIMERE 2016, through the OASIS3-MCT coupler"

_Geoscientific Model Development, 2016_

## Short Comment (SC1) · 28 Apr 2016

Dear authors,

In my role as Executive editor of GMD, I would like to bring to your attention our Editorial version 1.1:

http://www.geosci-model-dev.net/8/3487/2015/gmd-8-3487-2015.html

This highlights some requirements of papers published in GMD, which is also available on the GMD website in the 'Manuscript Types' section:

[Discussion paper]

[Figure]

http://www.geoscientific-model-development.net/submission/manuscript_types.html

In particular, please note that for your paper, the following requirements have not been met in the Discussions paper:

- "The main paper must give the model name and version number (or other unique identifier) in the title."

- "If the model development relates to a single model then the model name and the version number must be included in the title of the paper. If the main intention of an article is to make a general (i.e. model independent) statement about the usefulness of a new development, but the usefulness is shown with the help of one specific model, the model name and version number must be stated in the title. The title could have a form such as, "Title outlining amazing generic advance: a case study with Model XXX (version Y)"."

- "All papers must include a section, at the end of the paper, entitled 'Code availability'. Here, either instructions for obtaining the code, or the reasons why the code is not available should be clearly stated. It is preferred for the code to be uploaded as a supplement or to be made available at a data repository with an associated DOI (digital object identifier) for the exact model version described in the paper. Alternatively, for established models, there may be an existing means of accessing the code through a particular system. In this case, there must exist a means of permanently accessing the precise model version described in the paper. In some cases, authors may prefer to put models on their own website, or to act as a point of contact for obtaining the code. Given the impermanence of websites and email addresses, this is not encouraged, and authors should consider improving the availability with a more permanent arrangement. After the paper is accepted the model archive should be updated to include a link to the GMD paper."

- Inclusion of Code and/or data availability sections is mandatory for all papers and should be located at the end of the article, after the conclusions, and before any appendices or acknowledgments. For more details refer to the code and data policy.

Upon revision of the article, please add the version numbers of CHIMERE and WRF used for this publication to the title. Additionally, the code availability section should contain a statement about the WRF model as well.

Yours,

Astrid Kerkweg

---

## Referee Comment (RC1) · Anonymous Referee #1 · 13 Jul 2016

This is an interesting paper and it reflects the fact of more recognition that coupling meteorological model with chemistry transport model with online feedback is an important way to make the modeling system having a more realistic representation. Overall the paper was well written and well organized. However, it lacks of one crucial information, in my opinion, and that makes the paper less convincing. That piece of information is the location of the aerosol, in particular, mineral dust. Authors have pointed out the relationship of aerosol direct and semi-direct effect and radiative budget. It will be great if authors can show where the mineral dust is. With this information, it will be much clearer and convincing in determining the performance of the model with respect to surface temperature and short wave surface radiation. Another thing is AOD calculation is part of CHIMERE model (the offline version) and it was included in the performance evaluation of the online model but did not do a sufficient analysis to explain why the online model did not do anything. The authors just explained it could be due to more resolved meteorology and change of wind speed. I believe this paper requires major revision before any consideration. Here are the detailed comment/question:

* it might be nice (more informative) to indicate usage of OASIS in the title

* page 4, line 8 - 11, what compiling flags? Why hard-coded? Should that be considered a bad coding practice?

* page 4, line 20, why it is impossible for OASIS to perform spatial interpolation? Fortran (I assume that what CHIMERE is written with) is a column major language so when collapsing a 3D array to a 1D array, it is quite easy to figure out where the spatial boundary in the 1D array.

* page 4, does CHIMERE use the same domain decomposition method as in WRF?

* paper 5, section 2.3.2, should you mention what LW and SW schemes are being used in WRF? RRTMG was mentioned in the later portion of the paper. Also what are the chemical species are being used for the optics calculation?

* page 6, based on Figure 1, it seems to me (I don't know CHIMERE well) that the online model will make met data available to CHIMERE at time step 0, 600, 1200 and so forth. So within CHIMERE at time step 300, 900 and so forth, which time step meta data is being used?

* page 10, what is LUCIA?

* page 11, Why there is a big gap (close to 200 seconds) between case 2 and 1 when 1 core was used? Why not show the actual model run time rather than average per iteration (were there any big variation in run time among different iteration)?

* page 12, Why there is a big and gradual increase when number of cores > 12? In general, why case 2 used more time that case 1? What constitute that difference?

* page 15, Figure information such as number and title, is missing.

* page 15, 16, figures missing label on the y-axis on the right hand side.

* page 17, figure missing label on the y-axis. Is the GMT time or local time being used in the graph?

* page 17, Why case 2's temp increased during the second half of day time period on every day?

* page 33, why the number of data points in each site is so small in a 44 days period?

---

## Referee Comment (RC2) · Anonymous Referee #2 · 22 Jul 2016

General Comments:

This paper describes the development of an on-line coupled chemistry-weather model using the WRF weather forecasting model and CHIMERE chemical transport model, coupled for the first time using OASIS –MCT. The authors then assess the impact of the coupled system on the regional simulation of aerosols over Northern Africa and Europe and the feedback of the aerosols due to the direct and semi-direct effect on model radiation fields, surface temperature and wind fields. Observations of aerosol optical depth and temperature profiles are used to evaluate the performance of the

three different test cases (offline, online coupling of meteorological fields to CHIMERE only, and online coupling of met fields and aerosol between CHIMERE and WRF).

While the development of the online coupled chemistry-weather models and impact on weather forecasts is of increasing interest in both the weather and atmospheric composition communities, I find this paper poorly written and lacking in a substantial discussion of what are the principle aims of the paper. Evaluating the impacts of the inclusion of aerosol-radiation-interactions on the radiation balance in the model and other meteorological fields is of interest to the community but it is not new, particularly when the authors focus on a dust specific case and make no mention of the role of anthropogenic emissions, even though CHIMERE simulates more than just dust aerosol. The current evaluation does nothing but confirm the findings of many other dust specific studies already published in the literature (not all of which are referenced here). What the paper is crucially lacking is a discussion on what are the potential benefits (or not) of having a fully online coupled chemistry-weather system for (a) aerosol performance (as assessed through Online Case 1) and (b) weather forecast skill (as assessed in Online Case 2) versus the offline model. There is no clear statement on these points. The paper would strongly benefit from a Discussion section summarizing the results and putting them into context of the aims of the paper. This is highlighted by the authors use of language such as "may result" a number of times (such as in the discussion of temperature impacts and AOD). There should be no ambiguity in the results and the authors should be able to clearly demonstrate their findings with confidence in order to draw conclusions.

I would not publish this paper in this current form but it would be publishable if substantial improvements were made in terms of both presentation and scientific content. Major revisions are requested as outlined below.

Specific Comments:

Abstract, Line 3: "several distinct models are involved" . Do you mean parameteriza-

tions? It would be more correct to highlight the inherent uncertainty in the processes involved.

Abstract, line 7: "This is mainly due to some additional computations made within the models such as more frequent calls to meteorology..." – what about the additional cost due to the additional number of tracers in the CTM? How significant is this cost?

Paper Layout: I would recommend putting the model descriptions described in Section 3 before the Coupling description section. It is very difficult to follow the coupling description without some knowledge of the individual models.

I would also strongly recommend the inclusion of a Discussion section before the Conclusions to bring the evaluation presented in Section 5 into context in terms of the main aims of the paper.

The main aims of the paper should be clearly stated in the Introduction.

Introduction: There are much more appropriate references for the direct and indirect aerosol effects, please update. Use of the word "effects" to describe aerosol feedback as opposed to "interactions" – CMIP5 did update this terminology for the direct and indirect effects to be aerosol-radiation interactions and aerosol-cloud-radiation interactions. I would recommend updating the terminology to what is now routinely used in the literature.

Introduction, page 2 line 2: The statement that aerosol effects are neglected by offline models, is this strictly true? Do these models not have a climatological representation of aerosols or even just use some fixed numbers? If so then they do not neglect them but just have a very (possibly overly) simplified approach to representing them.

Coupling Methods and Assessment of Computational Performance: How can you assess the impact of the coupled system on the computational performance of the model without appropriate load balancing? The results presented here are therefore not a correct representation of what an optimally load balanced system would look like, severely

affecting the WRF wait time for instance. It is well known that CTMs are very computationally intensive and require appropriate load balancing and I am surprised the authors have not done this. I find this a major flaw in the scientific methods and recommend redoing the computational analysis on an appropriately load balanced system.

The authors refer in a number of places (in the Abstract, Section 2.4 and Conclusions for instance) on the additional computational time requirements of CHIMERE "due to more frequent calls to meteorology treatment routines" when in ONLINE mode. Can the authors please expand on this ? What routines need to be called more frequently and why? Section 5: I would include in the section title that this is a dust specific case study.

In the WRF-CHIMERE online simulations covering the May to July 2012 period, is WRF free-running? I assume it is being driven by an analysis. If this is the please include details of how WRF is driven and how frequently. If it's not the case then a free-running model is very quickly going to diverge from the true meteorological conditions which will severely impact the simulation and distribution of aerosols and associated biases.

Impacts on radiation would benefit from a link to the spatial distribution of the dust. Inclusion of a spatial AOD plot in Figure 7 or dust AOD if possible would highlight the impacts better.

Figure 7: Are these all-sky fluxes? The increase in net SW at the surface is linked to cloud changes, is this just a surmise or have the authors evaluated this? For example, did the authors assess the impact on the SW cloud forcing? Please improve this discussion and wording.

End of Section 5.1 – what is the impact of the aerosol feedback on dust emission? A plot would be informative or even a statement of the regional % change.

Technical Corrections:

Figure 7 has no caption or figure labels

All figures are missing appropriate labels (a), (b), (c) etc.

Figure 11: it is much clearer if (model-obs) is plotted rather than (obs – model) , in the former a negative value is associated with a negative bias or underestimation which is much clearer than vice versa with the former.

Page 1, line 21: The aerosol effects processes –> ?? see my earlier comment on updating the terminology of aerosol effects –> interactions.

Page8, line16: Please include appropriate references for WFR and CHIMERE configurations.

Page 9, line 8: planeray –> planetary

Page 9, line 12 – put link in parentheses

Page 9, line 11: Anthropic –> Anthropogenic

Page 10, line 2: Tests consist in –> Tests consist of

Page 10, line 2: 64 cores computer –> 64 computer cores

Page 14: Line 12: The perturbation is dominated by dust "as observed" – what observations are you referring to here?

Page 18 line, second last line: Above 1000 meter each of the two simulations perform better alternatively – badly phrased sentence which doesn't make sense, reword.

Page 19, first paragraph and in other areas of the manuscript the authors use the phrase "difference among models" –> difference between models. Change all occurrences.

Section 5.4, last paragraph: There is no discussion here of the contribution of anthropogenic source to the AOD observed at particularly more northern European stations. There is very little difference between the 3 test cases really in terms of rmse and bias and correlations. Can the authors draw a conclusion on the role of the more highly

resolved meteorology on improving AOD simulations.

PM10 evaluation: a plot or a table summarizing the PM10 results would be useful. The authors should acknowledge that there is a lot more at play here than just the online coupling, in terms of aerosol size distribution, transport and removal processes of in particular dust and sea salt will play a large role in PM10.

---

## Author Comment (AC1) · 29 Sep 2016

**Answers to executive editor comment and referees for "Aerosol effects modeling using an online coupling between the meteorological model WRF and the chemistry-transport model CHIMERE " by Briant et al. (gmd-2016-73)**

Dear Editor and reviewers,

Please find in this letter our answers to the comments made during the review process. Our answers are in blue in the text and after each remark. The article was fully revised following the recommendations. In particular, the following issues have been addressed :

— The code availability section is now complete
— The location of mineral dust sources is now shown and discussed
— The aim of the study is clearly stated
— Several paragraph have been reworded in order to clarify our statements.

We acknowledge the executive editor and the two reviewers for their comments.

**1 Executive editor**

**A. Kerkweg**

kerkweg@uni-mainz.de

Received and published : 28 April 2016

Dear authors, In my role as Executive editor of GMD, I would like to bring to your attention our Editorial version 1.1 :

`http://www.geosci-model-dev.net/8/3487/2015/gmd-8-3487-2015.html`

This highlights some requirements of papers published in GMD, which is also available on the GMD website in the Manuscript Types section :

`http://www.geoscientific-model-development.net/submission/manuscript_types.html`

In particular, please note that for your paper, the following requirements have not been met in the Discussions paper :

— "The main paper must give the model name and version number (or other unique identifier) in the title."
— If the model development relates to a single model then the model name and the version number must be included in the title of the paper. If the main intention of an article is to make a general (i.e. model independent) statement about the usefulness of a new development, but the usefulness is shown with the help of one specific model, the model name and version number must be stated in the title. The title could have a form such as, Title outlining amazing generic advance : a case study with Model XXX (version Y).
— "All papers must include a section, at the end of the paper, entitled Code availability. Here, either instructions for obtaining the code, or the reasons why the code is not available should be clearly stated. It is preferred for the code to be uploaded as a supplement or to be made available at a data repository with an associated DOI (digital object identifier) for the exact model version described in the paper. Alternatively, for established models, there may be an existing means of accessing the code through a particular system. In this case, there must exist a means of permanently accessing the precise model version described in the paper. In some cases, authors may prefer to put models on their own website, or to act as a point of contact for obtaining the code. Given the impermanence of websites and email addresses, this is not encouraged, and authors should consider improving the availability

with a more permanent arrangement. After the paper is accepted the model archive should be updated to include a link to the GMD paper."
— Inclusion of Code and/or data availability sections is mandatory for all papers and should be located at the end of the article, after the conclusions, and before any appendices or acknowledgments. For more details refer to the code and data policy.

Upon revision of the article, please add the version numbers of CHIMERE and WRF used for this publication to the title. Additionally, the code availability section should contain a statement about the WRF model as well.

CHIMERE, WRF and OASIS version numbers were added in the revised paper title. In addition, the 'Code availability section' was updated. It now contains instructions for obtaining both WRF and CHIMERE model codes along with the OASIS coupler.

**2    Referee #1 :**

Received and published : 13 July 2016

This is an interesting paper and it reflects the fact of more recognition that coupling meteorological model with chemistry transport model with online feedback is an important way to make the modeling system having a more realistic representation. Overall the paper was well written and well organized. However, it lacks of one crucial information, in my opinion, and that makes the paper less convincing. That piece of information is the location of the aerosol, in particular, mineral dust. Authors have pointed out the relationship of aerosol direct and semi-direct effect and radiative budget. It will be great if authors can show where the mineral dust is. With this information, it will be much clearer and convincing in determining the performance of the model with respect to surface temperature and short wave surface radiation.

A figure showing the location of mineral dust sources along with an explanation paragraph has been added in the revised version.

Another thing is AOD calculation is part of CHIMERE model (the offline version) and it was included in the performance evaluation of the online model but did not do a sufficient analysis to explain why the online model did not do anything. The authors just explained it could be due to more resolved meteorology and change of wind speed. I believe this paper requires major revision before any consideration.

A more detailed explanation has been added in the revised version.

Here are the detailed comment/question :

* it might be nice (more informative) to indicate usage of OASIS in the title
OASIS3-MCT has been added in the revised version of the paper.

* page 4, line 8 - 11, what compiling flags ? Why hard-coded ? Should that be considered a bad coding practice ?
The compilation flags are "cpl_wrf_chimere" and "cpl_wrf_chimere". We added it in the revised version of the paper.

Hard-coded material should not be considered as bad coding practice in general. In our case the NEMO variable names are hard coded within the WRF-OASIS interface, within the WRF model code. This is already included within WRF released and is widely distributed. However, the

hard-coded variable names, makes it impossible for us to re-use the already existing WRF-OASIS interface for the WRF-CHIMERE coupling. We could have wrote our own WRF-OASIS interface, but instead we preferred to separate the WRF-OASIS interface from the NEMO related material, using compilation flags, in order to be able to re-use the existing interface. Therefore, we developed the WRF-CHIMERE coupling in a way that it does not interfere with the already existing, and already distributed, on-line coupling. We reword the paragraph in the revised version to better explain this.

* page 4, line 20, why it is impossible for OASIS to perform spatial interpolation ? Fortran (I assume that what CHIMERE is written with) is a column major language so when collapsing a 3D array to a 1D array, it is quite easy to figure out where the spatial boundary in the 1D array. CHIMERE is indeed written in Fortran. We agree that it would be possible to perform a spatial interpolation with the 3D arrays collapsed into 1D array. However, OASIS is designed to performed spatial interpolation with 2D arrays only. But, in fact, the way to exchange the data is not a problem. Considering that we know how OASIS works, this is just a development to adapt the format. This has strictly no impact on the results. We reword the paragraph in the revised manuscript to better explain this.

* page 4, does CHIMERE use the same domain decomposition method as in WRF ? CHIMERE and WRF do not use the same decomposition method. This is stated on page 4, line 24-25.

* paper 5, section 2.3.2, should you mention what LW and SW schemes are being used in WRF ? RRTMG was mentioned in the later portion of the paper. Also what are the chemical species are being used for the optics calculation ? The RRTMG scheme is now mentionned in Section 2.3.2. Aerosol species are used for the optics calculation.

* page 6, based on Figure 1, it seems to me (I dont know CHIMERE well) that the online model will make met data available to CHIMERE at time step 0, 600, 1200 and so forth. So within CHIMERE at time step 300, 900 and so forth, which time step meta data is being used ? On Figure 1, WRF meteorological field are sent to CHIMERE every 600 seconds, while aerosol optical properties are sent from CHIMERE to WRF every 1800 seconds. On the other hand both WRF and CHIMERE may perform sub-iterations, in order to meet the CFL condition for instance. In that case the previously received data are used during the sub-iterations. We re-word the explanation in the revised version of the paper.

* page 10, what is LUCIA ? LUCIA means : Load-balancing Utility and Coupling Implementation Appraisal. We added them to the revised paper.

* page 11, Why there is a big gap (close to 200 seconds) between case 2 and 1 when 1 core was used ? The increase of computational time between case 2 and case 1, when reducing the number of core, is not linear. Indeed, the amount of allocated data along with the number of operation to perform on each core do not increases linearly when reducing the number of core. Therefore explaining the 200 seconds gap when 1 core was used.

Why not show the actual model run time rather than average per iteration (were there any big variation in run time among different iteration) ?
Yes, some discrepancies among iterations may occur and using the average time per iteration instead of the total model run allow to reduce the measurements uncertainties.

* page 12, Why there is a big and gradual increase when number of cores ¿ 12 ?
The increase is due to the increase of the amount of data exchanged among the cores. When increasing the number of cores each model has to communicate more data to the other cores. On the other hand, this is balanced by the fact that each core has less calculations to perform, as its sub-domain is smaller.

In general, why case 2 used more time that case 1 ? What constitute that difference ?
In case 2 CHIMERE computes the optical properties, which is not done in case 1. This is responsible for the most part of the time increase. This is stated on page 10, line 27-28.

* page 15, Figure information such as number and title, is missing.
This is a layout issue that is corrected in the revised version of the paper. Figures on the page 15 are the Figure 7 :
Difference in WRF radiation budget at the ground surface between both online simulations for long-wave (top), short-wave (middle) and the sum of long-wave and short-wave (bottom). Fluxes are in W.m$^{-2}$ and are averaged in time over the period ranging from the 1$^{st}$ of June to the 14$^{th}$ of July.

* page 15, 16, figures missing label on the y-axis on the right hand side.
Y-axis labels are displayed on the left hand side. On the right hand this is the colorbar with values having units explained in the caption.

* page 17, figure missing label on the y-axis. Is the GMT time or local time being used in the graph ?
It is GMT time. This has been stated in the revised version of the paper.

* page 17, Why case 2s temp increased during the second half of day time period on every day ?
The increase is due to case 2's aerosol optical properties feedback. Indeed, the short-wave effect prevails during daytime, thus creating a decrease of the surface temperatures, while the long-wave effect alone contributes at night due to the earth outgoing long-wave radiations, inducing an increase of the temperatures. This is stated on page 14, line 32 and page 16 line 1-2.

* page 33, why the number of data points in each site is so small in a 44 days period ?
AERONET time-series are rarely complete. The level2 data used in this study correspond to the clear-sky product, i.e the data where no cloud was identified. The AOD is thus only representative of aerosol.

**3   Referee #2 :**

Received and published : 22 July 2016

General Comments :
This paper describes the development of an on-line coupled chemistry-weather model using the WRF weather forecasting model and CHIMERE chemical transport model, coupled for the first

time using OASIS MCT. The authors then assess the impact of the coupled system on the regional simulation of aerosols over Northern Africa and Europe and the feedback of the aerosols due to the direct and semi-direct effect on model radiation fields, surface temperature and wind fields. Observations of aerosol optical depth and temperature profiles are used to evaluate the performance of the three different test cases (offline, online coupling of meteorological fields to CHIMERE only, and online coupling of met fields and aerosol between CHIMERE and WRF).

While the development of the online coupled chemistry-weather models and impact on weather forecasts is of increasing interest in both the weather and atmospheric composition communities, I find this paper poorly written and lacking in a substantial discussion of what are the principle aims of the paper. Evaluating the impacts of the inclusion of aerosol-radiation-interactions on the radiation balance in the model and other meteorological fields is of interest to the community but it is not new, particularly when the authors focus on a dust specific case and make no mention of the role of anthropogenic emissions, even though CHIMERE simulates more than just dust aerosol. The current evaluation does nothing but confirm the findings of many other dust specific studies already published in the literature (not all of which are referenced here). What the paper is crucially lacking is a discussion on what are the potential benefits (or not) of having a fully online coupled chemistry-weather system for (a) aerosol performance (as assessed through Online Case 1) and (b) weather forecast skill (as assessed in Online Case 2) versus the offline model. There is no clear statement on these points. The paper would strongly benefit from a Discussion section summarizing the results and putting them into context of the aims of the paper. This is highlighted by the authors use of language such as may result a number of times (such as in the discussion of temperature impacts and AOD). There should be no ambiguity in the results and the authors should be able to clearly demonstrate their findings with confidence in order to draw conclusions. I would not publish this paper in this current form but it would be publishable if substantial improvements were made in terms of both presentation and scientific content. Major revisions are requested as outlined below.

Specific Comments :
Abstract, Line 3 : several distinct models are involved . Do you mean parameterizations ? It would be more correct to highlight the inherent uncertainty in the processes involved.
The cited models correspond to the meteorological and the chemistry transport models. The models are used as it. The only new thing in this study is to add a dialog between the two, but without any change in the parameterizations of each one. For example, WRF already uses aerosols concentrations values to estimate a contribution of aerosol on radiation. The coupling presented in this article does not change the WRF calculation. But, the aerosol concentrations shift from 'climatological' (the current ay with WRF) to 'sub-hourly' with the forcing proposed by the CHIMERE simulation.

Abstract, line 7 : This is mainly due to some additional computations made within the models such as more frequent calls to meteorology... what about the additional cost due to the additional number of tracers in the CTM ? How significant is this cost ?
As answered for the previous question, the models were not changed. They preserve their own schemes and parameterizations. The CHIMERE model simulates tens of gaseous and aerosol chemical species and this is conserved. There is no tracers in our modelling, the model being dedicated to simulate realistic chemical composition of the atmosphere.

Paper Layout : I would recommend putting the model descriptions described in Section 3 before the Coupling description section. It is very difficult to follow the coupling description without some knowledge of the individual models.

Section 2 describe the new developments that were made while Section 3 described specific model configurations that are used in the later simulation. We believe that describing the specifics of the simulations before describing the what we actually have done along with what we want to assess would not be relevant. For instance, it would make no sense to talk about case 1 or case 2 simulations before explaining what is acually exchanged through the coupler. In addition, the reading of each model reference papers that are cited are appropriate and shall give sufficient information to prevent any lack of knowledge on the discussed models.

I would also strongly recommend the inclusion of a Discussion section before the Conclusions to bring the evaluation presented in Section 5 into context in terms of the main aims of the paper. The discussion made within Section 5 is brough back to the paper aims within within the conclusion.

The main aims of the paper should be clearly stated in the Introduction. We reworded the introduction in the revised version of the paper to emphasize more the paper aims.

Introduction : There are much more appropriate references for the direct and indirect aerosol effects, please update. Use of the word effects to describe aerosol feedback as opposed to interactions CMIP5 did update this terminology for the direct and indirect effects to be aerosol-radiation interactions and aerosol-cloud-radiation interactions. I would recommend updating the terminology to what is now routinely used in the literature. We did our best for the bibliography. It seems that our bibliography is not the best for the reviewer, but unfortunately, he didn't suggest new reference. Thus, we didn't really understand what was missing in our text. We updated the terminology in the revised version of the paper.

Introduction, page 2 line 2 : The statement that aerosol effects are neglected by offline models, is this strictly true ? Do these models not have a climatological representation of aerosols or even just use some fixed numbers ? If so then they do not neglect them but just have a very (possibly overly) simplified approach to representing them. Yes, these models do have a climatological representation of aerosols. We reworded the sentence in the revised version of the paper, to specify that these model use a simplified approach to represent the aerosol effects.

Coupling Methods and Assessment of Computational Performance : How can you assess the impact of the coupled system on the computational performance of the model without appropriate load balancing ? The results presented here are therefore not a correct representation of what an optimally load balanced system would look like, severely affecting the WRF wait time for instance. It is well known that CTMs are very computationally intensive and require appropriate load balancing and I am surprised the authors have not done this. I find this a major flaw in the scientific methods and recommend redoing the computational analysis on an appropriately load balanced system. The load balance may depend on various parameter such as the parameterization that are used, the number of species that are used, the number of cores that are available. Giving one specific ratio would not be fair as it would not always be applicable.

Instead, we decided to evaluate the Computational performance with an equal number of core for both models. The results are not optimal but the drawn conclusions are still valid. In addition, it is stated (Section 4.3) that iterative methods have to be used to the number of cores to use with each model that will optimize to computational burden. We reworded the Section 4.3 to better explain

this in the revised version of the paper.

The authors refer in a number of places (in the Abstract, Section 2.4 and Conclusions for instance) on the additional computational time requirements of CHIMERE due to more frequent calls to meteorology treatment routines when in ONLINE mode. Can the authors please expand on this? What routines need to be called more frequently and why?
In offline mode CHIMERE read WRF meteorology files every hour. In online mode CHIMERE receive WRF meteorology, through OASIS, at a higher rate (i.e. 15 mins in Section 4 and 5 simulations). Therefore, CHIMERE needs to process WRF meteorology more often in online mode. We added this explanation in the revised version of the paper.

Section 5 : I would include in the section title that this is a dust specific case study.
We changed the section title in the revised version of the paper.

In the WRF-CHIMERE online simulations covering the May to July 2012 period, is WRF free-running? I assume it is being driven by an analysis. If this is the please include details of how WRF is driven and how frequently. If its not the case then a free-running is very quickly going to diverge from the true meteorological conditions which severely impact the simulation and distribution of aerosols and associated biases.
As stated in Section 3.1 : WRF is forced by the meteorological analysis data of NCEP/GFS (Kalnay et al., 1996) provided on a regular $1.125° \times 1.125°$ grid. The forcing frequency is 3 hours. This has been added to the revised version of the paper.

Impacts on radiation would benefit from a link to the spatial distribution of the dust. Inclusion of a spatial AOD plot in Figure 7 or dust AOD if possible would highlight the impacts better.
A figure showing the location of mineral dust sources along with an explanation paragraph has been added in the revised version in order to link the impact on radiation with the mineral dust emissions.

Figure 7 : Are these all-sky fluxes? The increase in net SW at the surface is linked to cloud changes, is this just a surmise or have the authors evaluated this? For example, did the authors assess the impact on the SW cloud forcing? Please improve this discussion and wording.
Yes, these are all-sky fluxes. We improved the discussion and wording in the revised version of the paper.

End of Section 5.1  what is the impact of the aerosol feedback on dust emission? A plot would be informative or even a statement of the regional % change.
We added a statement of the % of mineral dust emission change in the revised version of the paper.

Technical Corrections :
Figure 7 has no caption or figure labels
This is a layout issue that is corrected in the revised version of the paper. Figures on the page 15 are the Figure 7 :
Difference in WRF radiation budget at the ground surface between both online simulations for long-wave (top), short-wave (middle) and the sum of long-wave and short-wave (bottom). Fluxes are in $W.m^{-2}$ and are averaged in time over the period ranging from the $1^{st}$ of June to the $14^{th}$ of July.

All figures are missing appropriate labels (a), (b), (c) etc.

This has been corrected in the revised version of the paper.

Figure 11 : it is much clearer if (model-obs) is plotted rather than (obs  model) , in the former a negative value is associated with a negative bias or underestimation which is much clearer than vice versa with the former.
This has been corrected in the revised version of the paper.

Page 1, line 21 : The aerosol effects processes ¿ ? ? see my earlier comment on updating the terminology of aerosol effects ¿ interactions.
We updated the terminology in the revised version of the paper.

Page8, line16 : Please include appropriate references for WFR and CHIMERE configurations.
CHIMERE 2016 new reference has been added in the revised version of the paper. However there is no WRF reference, other than what is already cited. In addition, WRF and CHIMERE configurations are fully described in Sections 3.1 and 3.2.

Page 9, line 8 : planeray ¿ planetary
This has been corrected in the revised version of the paper.

Page 9, line 12  put link in parentheses
This has been corrected in the revised version of the paper.

Page 9, line 11 : Anthropic ¿ Anthropogenic
This has been corrected in the revised version of the paper.

Page 10, line 2 : Tests consist in ¿ Tests consist of
This has been corrected in the revised version of the paper.

Page 10, line 2 : 64 cores computer ¿ 64 computer cores
This has been corrected in the revised version of the paper.

Page 14 : Line 12 : The perturbation is dominated by dust as observed  what observations are you referring to here ?
We are referring to Figure 7 changes. We reworded the sentence in the revised version of the paper.

Page 18 line, second last line : Above 1000 meter each of the two simulations perform better alternatively  badly phrased sentence which doesnt make sense, reword.
We reworded the sentence in the revised version of the paper.

Page 19, first paragraph and in other areas of the manuscript the authors use the phrase difference among models ¿ difference between models. Change all occurrences.
We changed all occurences in the revised version of the paper.

Section 5.4, last paragraph : There is no discussion here of the contribution of anthropogenic source to the AOD observed at particularly more northern European stations. There is very little difference between the 3 test cases really in terms of rmse and bias and correlations. Can the authors draw a conclusion on the role of the more highly resolved meteorology on improving AOD simulations.
We added discussion of the contribution of anthropogenic source to the AOD in the revised version

PM10 evaluation : a plot or a table summarizing the PM10 results would be useful. The authors should acknowledge that there is a lot more at play here than just the online coupling, in terms of aerosol size distribution, transport and removal processes of in particular dust and sea salt will play a large role in PM10.

We believe that a plot or a table is not relevant here as it would not add any more information than the averaged information. However, we do acknowledge that Section 5.5 alone is not enough to properly assess the aerosol effect impact on PM10 concentrations. Nonetheless, the paper focus is on the presentation of online coupling, with a first evaluation that is not exhaustive. More study will be made in the future that will adress the PM10 concentrations topic with more details. We emphasized that in the revised version of the paper.

---

## Author Response (AR2)

**Answers to referee #1 report for "Aerosol-radiation interactions modeling using an online coupling between the meteorological model WRF 3.7.1 and the chemistry-transport model CHIMERE 2016, through the OASIS3-MCT coupler" by Briant et al. (gmd-2016-73)**

Dear Editor and reviewers,

This paper describes with technical details a new online coupling between two models. In addition, an initial evaluation of this coupling is made. We believe that it is of scientific interest and lies within the scope of the Geoscientific Model Development journal.

Please find in this letter our answers to the anonymous referee #1 comments. We understand the reviewer concerns and we attempt here to answer for the best. Our answers are in blue in the text and after each remark. We acknowledge the executive editor and the two reviewers for their comments.

Submitted on 29 Oct 2016
Anonymous Referee #1

**Referee #1 report**

**Suggestions for revision or reasons for rejection** (will be published if the paper is accepted for final publication)
The model might have weakness to represent the short-wave aerosol direct effect. In Figure 10, even long wave has been included, the net reduction in surface radiation did not result in reduction of surface temperature.

We observe a temperature increase over the Saharan region, the Arabian peninsula and off the western Saharan coast, where the radiative forcing (short wave + long wave) is positive. On the other hand, we observe a temperature decrease over Europe and the Sub-Saharan Africa, where the radiative forcing is negative. Figure 10 illustrates the highest increase of surface temperature that we observed over the domain. This is stated and discussed in the fourth and fifth paragraphs of Section 5.1. We reworded those paragraphs in the revised version.

The authors should use the model produced aerosol information, which is transferred to the meteorological model, to establish the relationship between the presence of aerosol, surface radiation, surface temperature, and AOD. However, the authors chose to use dust emission information as the base. This might not be strong enough since a CTM will calculate concentration of various chemical species through transport processes, chemistry and aerosol thermodynamics.

The mineral dust emission information should not be considered as "the base". Indeed, it is used in Section 5.1 to illustrate the following statement :
*"the perturbation in the optical properties is dominated by mineral dust, as observed changes occur over regions where mineral dust constitute the main aerosol type (i.e. Saharan region and Arabian peninsula)."*

On the other hand, we did use the aerosol information (i.e. the aerosol optical depth) that is transferred to the meteorological model. Indeed, in Section 5.3 and 5.4 modeled data is compared to

aerosol optical depth measurements.

The organization of the paper needs some more work. For instance, Fig 8 which is the origin of the story if you will, should be placed before Fig 7.

In the first paragraph of Section 5.1, we introduced radiation budget difference maps (Figure 7). We then we tried to explain the observed differences in the second paragraph. The hypothesis is that the observed differences are mainly due to mineral dust. Figure 8 is there to support that hypothesis. Consequently, the figures order is appropriate.

Figure 10, align part a and b so reader can easier to follow the relationship between temperature and radiation.

We aligned Figure 10 part a and b in the revised version.

In the fourth response, it was indicated the domain decompositions are different but in the paper, it says, it may be different (on page 4, line 26).

Indeed, domain decompositions may be different and, in most cases, they are. In offline mode, there is no reason that would force WRF and CHIMERE to have the same domain decomposition. When we developed the coupling, we could have added such a constraint for the online mode. This would have simplified the OASIS partition. Instead, we choose not to add this constraint for the simple reason that it can be avoided by the use of OASIS "points" partition. Therefore, we implemented the coupling in a way that the domain decomposition may be different for both models (as it is the case in offline mode). This is explained in Section 2.2, line 26-30. We added some additional explanation in the revised version.

I am not still quite satisfy with response 9 regarding to the gradual increase when number of cores > 12.

[revised manuscript text omitted]
 | 20 | 0.2 | 0.23 | 0.25 | 0.24 | 0.17 | 0.18 | 0.19 | 0.8 | 0.8 | 0.77 | 0.03 | 0.04 | 0.04 |